

# SABMIS: sparse approximation based blind multi-image steganography scheme

Rohit Agrawal[1,2], Kapil Ahuja[1], Marc C. Steinbach[3] and Thomas Wick[3]

[1] Computer Science and Engineering, Indian Institute of Technology Indore, Indore, India
[2] School of Computer Science Engineering and Technology, Bennett University, Greater Noida, India
[3] Leibniz Universität Hannover, Institut für Angewandte Mathematik, Hannover, Germany

## ABSTRACT

We hide grayscale secret images into a grayscale cover image, which is considered to be a challenging steganography problem. Our goal is to develop a steganography scheme with enhanced embedding capacity while preserving the visual quality of the stego-image as well as the extracted secret image, and ensuring that the stego-image is resistant to steganographic attacks. The novel embedding rule of our scheme helps to hide secret image sparse coefficients into the oversampled cover image sparse coefficients in a staggered manner. The stego-image is constructed by using the Alternating Direction Method of Multipliers (ADMM) to solve the Least Absolute Shrinkage and Selection Operator (LASSO) formulation of the underlying minimization problem. Finally, the secret images are extracted from the constructed stego-image using the reverse of our embedding rule. Using these components together, to achieve the above mentioned competing goals, forms our most novel contribution. We term our scheme SABMIS (Sparse Approximation Blind Multi-Image Steganography). We perform extensive experiments on several standard images. By choosing the size of the length and the width of the secret images to be half of the length and the width of cover image, respectively, we obtain embedding capacities of 2 bpp (bits per pixel), 4 bpp, 6 bpp, and 8 bpp while embedding one, two, three, and four secret images, respectively. Our focus is on hiding multiple secret images. For the case of hiding two and three secret images, our embedding capacities are higher than all the embedding capacities obtained in the literature until now (3 times and 6 times than the existing best, respectively). For the case of hiding four secret images, although our capacity is slightly lower than one work (about 2/3rd), we do better on the other two goals (quality of stego-image & extracted secret image as well as resistance to steganographic attacks). For our experiments, there is very little deterioration in the quality of the stego-images as compared to their corresponding cover images. Like all other competing works, this is supported visually as well as over 30 dB of Peak Signal-to-Noise Ratio (PSNR) values. The good quality of the stego-images is further validated by multiple numerical measures. None of the existing works perform this exhaustive validation. When using SABMIS, the quality of the extracted secret images is almost same as that of the corresponding original secret images. This aspect is also not demonstrated in all competing literature. SABMIS further improves the security of the inherently steganographic attack resistant transform based schemes. Thus, it is one of the most secure schemes among the existing ones. Additionally, we demonstrate that SABMIS executes in few minutes, and show its application on the real-life problems of securely transmitting medical images over the internet.

Corresponding author
Kapil Ahuja, kahuja@iiti.ac.in

# INTRODUCTION

The primary concern during the transmission of digital data over communication media is that anybody can access this data. Hence, to protect the data from being accessed by illegitimate users, the sender must employ some security mechanisms. In general, there are two main approaches used to protect secret data; cryptography (*Stallings, 2019*) and steganography (*Kordov & Zhelezov, 2021*), with our focus on the latter. Steganography is derived from the Greek words *steganos* for "covered" or "secret" and *graphie* for "writing". In steganography, the secret data is hidden in some unsuspected cover media so that it is visually imperceptible. Here, both the secret data as well as the cover media may be text or multimedia. Recently, steganography schemes that use images (binary, grayscale or color) as secret data as well as cover media have gained a lot of research interest due to their heavy use in World Wide Web applications. This is the *first* focus of our work.[1] Some real-life applications of this include securing biometric data, digital signature, personal banking information, and medical data.

Next, we present some relevant previous studies in this domain. Secret data can be hidden in images in two ways; spatially or by using a transform. In the spatial domain based image steganography scheme, secret data is hidden directly into the image by some modification in the values of the image pixels. These approaches have the drawback that they are inherently not resistant to steganographic attacks (*Artiemjew & Aleksandra, 2020*; *Hassaballah et al., 2021*). Some of the past works related to this are given in Table 1. The papers in this table are listed in the increasing order of the number of secret images hidden in the cover image.

In the transform domain based image steganography scheme, first, the image is transformed into frequency components, and then the secret data is hidden into these components. This process makes these approaches intrinsically resistant to steganographic attacks. Hence, such approaches form our *second* focus. Some of the past works related to this are given in Table 2. The papers in this table are listed in the increasing order of the number of secret images hidden in the cover image as well.

As mentioned above, images are of three kinds; binary, grayscale, and color. A grayscale image has more information than a binary image. Similarly, a color image has more information than a grayscale image. Thus, hiding a color secret image is more challenging than hiding a grayscale secret image, which is more challenging than hiding a binary secret image. Similarly, applying this concept to the cover image, we see a reverse sequence; see Table 3. We focus on the middle case here, *i.e.,* when both the secret images and the cover image are grayscale, which is considered challenging. This forms our *third* focus.

The difficulty in designing a good steganography scheme for hiding secret images into a cover image is increasing the embedding capacity of the scheme while preserving the quality of the resultant stego-image and extracted secret images as well as making the

[1] Hiding binary data into images is a different track, which we are not focusing in this article. For the sake of completeness, this is summarized in Appendix, 'Some steganography schemes for hiding binary secret data'.

**Table 1** Spatial domain-based image steganography schemes.

| Reference | Technique | Secret images | Cover image |
| --- | --- | --- | --- |
| *Baluja (2019)* | A modified version of Least Significant Bits (LSB) with deep neural networks | 2 color | color |
| *Gutub & Shaarani (2020)* | LSB | 2 color | color |
| *Guttikonda, Cherukuri & Mundukur (2018)* | LSB | 3 binary | grayscale and color |
| *Hu (2006)* | A modified version of LSB | 4 grayscale | grayscale |
| *Manujala & Danti (2015)* | A modified version of LSB | 4 color | color |

**Table 2** Transform domain-based image steganography schemes.

| Reference | Technique | Secret images | Cover image |
| --- | --- | --- | --- |
| *Sanjutha (2018)* | Discrete Wavelet Transformation (DWT) with Particle Swarm Optimization (PSO) | 1 grayscale | color |
| *Arunkumar et al. (2019a)* | Redundant Integer Wavelet Transform (RIWT) and QR Factorization | 1 grayscale | color |
| *Maheswari & Hemanth (2017)* | Contourlet and Fresnelet Transformations with Genetic Algorithm (GA) and PSO | 1 binary (specifically, QR code) | grayscale |
| *Arunkumar et al. (2019b)* | RIWT, Singular Value Decomposition (SVD) and Discrete Cosine Transformation (DCT) | 1 grayscale | grayscale |
| *Hemalatha et al. (2013)* | DWT | 2 grayscale | color |
| *Gutub & Shaarani (2020)* | DWT and SVD | 2 color | color |

**Table 3** Image types and levels of challenge.

| Image type | More challenging | Medium challenging | Less challenging |
| --- | --- | --- | --- |
| Secret image | Color | Grayscale | Binary |
| Cover image | Binary | Grayscale | Color |

scheme resistant to steganographic attacks. Hence, we need to balance these competing requirements. Here, not just the number of secret images but the total size of the secret images is also important. To capture this requirement of number as well as size, a metric of bits per pixel (bpp) is used.

In this work, we present a novel image steganography scheme wherein up to four images can be hidden in a single cover image. The size of the length and the width of a secret image is about half of the length and the width of the cover image, respectively, which results in a very high bpp capacity. No one has attempted hiding up to four secret images in a cover image with the transform domain based approach until now, and those who have attempted hiding one, or two images have also not achieved the level of embedding capacity that we

do. While enhancing the capacity as discussed above, the quality of our stego-image does not deteriorate much. Also, we do not need any cover image data to extract secret images on the receiver side, which is commonly required with other schemes. We do require some algorithmic settings on the receiver side, however, these can be communicated to the receiver separately. Thus, this makes our scheme more secure.

Let us consider the example of telediagnosis that refers to remote diagnosis. In this, medical images are distributed to some doctors for analyses and recommendations. During distribution, an unauthorized person can access these images and misuse them. To make this distribution process more secure, instead of directly sharing images, these can be hidden in a cover image using our steganography scheme and then the obtained stego-image can be shared. In this example, multiple secret images need to be shared (we consider sharing a maximum of four medical images). The existing transform based steganography schemes, which are inherently resistant to steganographic attacks, do not have such an embedding capacity. If we try to increase their capacity, then the quality of stego-image or extracted secret images gets degraded.

The most novel feature of our innovative scheme is that it is a combination of different components that helps us to achieve the competing goals of increasing embedding capacity, good quality stego-image as well as extracted secret images, and high resistance to steganographics attacks. Each of these components is discussed next.

The *first* component, *i.e.,* hiding of secret images, consists of the parts below.

(i) We perform sub-sampling on a cover image to obtain four sub-images of the cover image.

(ii) We perform block-wise sparsification of each of these four sub-images using DCT (Discrete Cosine Transform) and form respective vectors.

(iii) We represent each vector in two groups based upon large and small coefficients, and then oversample each of the resultant (or generated) sparse vector using a measurement matrix based linear measurements. The oversampling at this stage leads to a sparse approximation.

(iv) We repeat the second step above for each of the secret images.

(v) We embed DCT coefficients from the four secret images into "a set" of linear measurements obtained from the four sub-images of the cover image using our new embedding rule.

Amongst these parts, (i)–(ii) have been used in *Pal, Naik & Agrawal (2019)*; *Liu & Liao (2008)*; *Pan et al. (2015)* while (iii)–(v) are new.

*Second*, we generate the stego-image from these modified measurements by using the Alternating Direction Method of Multipliers (ADMM) to solve the Least Absolute Shrinkage and Selection Operator (LASSO) formulation of the underlying minimization problem. This method has a fast convergence, is easy to implement, and also is extensively used in image processing. Here, the optimization problem is an $\ell_1$-norm minimization problem, and the constraints comprise an *over-determined system of equations* (*Srinivas & Naidu, 2015*). Use of this component in steganography is first of its kind as well.

*Third*, we extract the secret images from the stego-image using our proposed extraction rule, which is the reverse of our embedding rule mentioned above. As mentioned earlier, we

do not require any information about the cover image while doing this extraction, which makes the process blind. Since our embedding procedure, as mentioned above, is new, thus the extraction part is also new. We call our scheme SABMIS (Sparse Approximation Blind Multi-Image Steganography), it is described in the section 'Proposed Approach'.

For performance evaluation, in the section 'Experimental Results' we perform extensive experiments on a set of standard images. We *first* compute the embedding capacity of our scheme, which turns out to be very good. *Next*, we check the quality of the stego-images by comparing them with their corresponding cover images. We use both a visual measure and a set of numerical measures for this comparison. The numerical measures used are: Peak Signal-to-Noise Ratio (PSNR) value, Mean Structural Similarity (MSSIM) index, Normalized Cross-Correlation (NCC) coefficient, entropy, and Normalized Absolute Error (NAE). The results show very little deterioration in the quality of the stego-images.

*Further*, we visually demonstrate the high quality of the extracted secret images by comparing them with the corresponding original secret images. *Also*, *via* experiments, we support our conjecture that our scheme is resistant to steganographic attacks. *Next*, we demonstrate efficiency of our scheme by providing timing data. *Finally*, we present application of our scheme on real-life data demonstrating its usefulness.

Also, we exhaustively compare SABMIS with competing schemes to demonstrate that it is among the best. For the sake of better exposition, this comparison is given in Introduction itself (see subsection below). Finally, in the section 'Conclusions and Future Work', we discuss conclusions and future work.

## Comparison with past work

Here, we predominately compare our SABMIS scheme with the existing steganography schemes for the embedding capacity, the quality of stego-images, and resistance to steganographic attacks. For the stego-image quality comparison, since most works have computed PSNR values only, we use only this metric for our analysis. Although we check the quality of the extracted secret images by comparing them with the corresponding original secret images (as earlier), this check is not common in the existing works. Hence, we do not perform this comparison.

In the literature, there exist multiple transform-based steganography schemes that hide one or two secret images. Hence, in Table 4 we compare our SABMIS scheme using the above mentioned metrics with such competing schemes. Recall, that like our SABMIS scheme these schemes are inherently resistant to steganographic attacks as well.

As evident from Table 4, for the case of hiding one secret image, we compare with the best work of this category (*Arunkumar et al., 2019b*). Here, as for us, by using a transform based approach, a grayscale secret image is hidden into a grayscale cover image. *Arunkumar et al. (2019b)* and our scheme both achieve an embedding capacity of 2 bpp. When comparing the stego-image and the corresponding cover image, *Arunkumar et al. (2019b)* achieve a PSNR value of 49.69 dB (when experimenting with eight cover images) while we achieve a lower PSNR value of 41.64 dB (when experimenting with a higher number of cover images, *i.e.*, ten). PSNR values over 30 dB are considered good (*Gutub & Shaarani, 2020*; *Zhang et al., 2013*; *Liu & Liao, 2008*). Although the scheme by *Arunkumar et al. (2019b)* is superior

**Table 4** Performance comparison of our SABMIS scheme with competing transform-based steganography schemes, which are inherently resistant to steganographic attacks.

| No. of secret images | Steganography scheme | Type of secret image | Type of cover images | EC (in bpp) | (Avg. PSNR, No. of cover images) | Max. PSNR | Resistant to steganographic attacks? |
|---|---|---|---|---|---|---|---|
| 1 | *Arunkumar et al. (2019b)* | Grayscale | Grayscale | 2 | (49.69, 8) | 50.15 | Yes |
| | SABMIS | Grayscale | Grayscale | 2 | (41.64, 10) | 46.25 | Yes |
| 2 | *Hemalatha et al. (2013)* | Grayscale | Color | 1.33 | (44.75, 2) | 44.80 | Yes |
| | SABMIS | Grayscale | Grayscale | 4 | (38.74, 10) | 42.60 | Yes |

than ours for hiding one secret image, it does not scale for the case of hiding multiple secret images, which we do (please see below).

For the case of hiding two secret images, we again compare with the best work of this category (*Hemalatha et al., 2013*). Here, using the transform based approach, two grayscale secret images are hidden into a color cover image. This setup is easier than our case where using a transform based approach, we embed two grayscale secret images into a grayscale cover image (see Table 3). *Hemalatha et al. (2013)* achieve an embedding capacity of 1.33 bpp while we achieve a higher embedding capacity of 4 bpp. When comparing the stego-image and the corresponding cover image, *Hemalatha et al. (2013)* achieve a PSNR value of 44.75 dB (when experimenting with only two cover images) while we achieve a lower PSNR value of 38.74 dB (when experimenting with a higher number of cover images, *i.e.,* ten). To sum-up, our scheme is better than the one by *Hemalatha et al. (2013)* because of the below reasons.

In terms of the quality of the scheme,

a) we target a harder problem than *Hemalatha et al. (2013)*, and

b) we achieve a higher embedding capacity than *Hemalatha et al. (2013)*.

In terms of the validation of the scheme,

a) we experiment with a large number of cover images (ten as compared to two in *Hemalatha et al. (2013)*),

b) as discussed earlier, we obtain PSNR values over 30 dB of stego-images, which are considered acceptable, and

c) we check the quality of stego-image on a greater number of numerical measures (five as compared to one in *Hemalatha et al. (2013)*).

When using the transform-based approach, no one has hidden three or four secret images in a cover image. To demonstrate the broad applicability of our scheme, in Table 5, we compare our SABMIS scheme using the above discussed metrics with the best spatial domain-based scheme that hide three and four secret images. Recall that, unlike our SABMIS scheme, these schemes are not intrinsically resistant to steganographic attacks. Please note that in the current scenario of transmitting stego-data over the internet, security is of paramount importance.

As evident from Table 5, for the case of hiding three secret images, we compare with the best work of this category (*Guttikonda, Cherukuri & Mundukur, 2018*). Here, three binary secret images are hidden into a grayscale cover image. As for the above case, this setup is

**Table 5** Performance comparison of our SABMIS scheme with competing spatial domain-based steganography schemes, which are not inherently resistant to steganographic attacks.

| No. of secret images | Steganography scheme | Type of secret image | Type of cover images | EC (in bpp) | (Avg. PSNR, No. of cover images) | Max. PSNR | Resistant to steganographic attacks? |
|---|---|---|---|---|---|---|---|
| 3 | *Guttikonda, Cherukuri & Mundukur (2018)* | Binary | Grayscale | 1 | (46.36, 2) | 46.38 | No |
| | SABMIS | Grayscale | Grayscale | 6 | (37.17, 10) | 41.06 | Yes |
| 4 | *Hu (2006)* | Grayscale | Grayscale | 12 | (34.80, 5) | 34.82 | No |
| | SABMIS | Grayscale | Grayscale | 8 | (35.66, 10) | 39.74 | Yes |

easier than our case of hiding three grayscale secret images into a grayscale cover image (again see Table 3). *Guttikonda, Cherukuri & Mundukur (2018)* achieve an embedding capacity of 1 bpp while we achieve a higher embedding capacity of 6 bpp. When comparing the stego-image and the corresponding cover image, *Guttikonda, Cherukuri & Mundukur (2018)* achieve a PSNR value of 46.36 dB (when experimenting with only two cover images) while we achieve a lower PSNR value of 37.17 dB (when experimenting with a higher number of cover images, *i.e.,* ten). To sum-up, our scheme is superior than the one by *Guttikonda, Cherukuri & Mundukur (2018)* because of the below reasons.

In terms of the quality of the scheme,

a) we target a harder problem than *Guttikonda, Cherukuri & Mundukur (2018)*,

b) we achieve a higher embedding capacity than *Guttikonda, Cherukuri & Mundukur (2018)*, and

c) we further improve the security of the inherently steganographic attack resistant transform based schemes.

In terms of the validation of the scheme,

a) we experiment with a large number of cover images (ten as compared to two in *Guttikonda, Cherukuri & Mundukur (2018)*),

b) as discussed earlier, we obtain PSNR values over 30 dB of stego-images, which are considered acceptable,

c) we check the quality of stego-image on a greater number of numerical measures (five as compared to one in *Guttikonda, Cherukuri & Mundukur (2018)*),

d) and we demonstrate the good quality of extracted secret images, which *Guttikonda, Cherukuri & Mundukur (2018)* do not.

Next, we compare with the best scheme that hides four secret images in a cover image, *i.e., Hu (2006)*. As for our case, all images (secret and cover) are grayscale. *Hu (2006)* achieve an embedding capacity of 12 bpp while we achieve a lower embedding capacity of 8 bpp. When comparing the stego-image and the corresponding cover image, *Hu (2006)* achieve a PSNR value of 34.80 dB (when experimenting with five cover images) while we achieve a higher PSNR value of 35.66 dB (when experimenting with a higher number of cover images, *i.e.,* ten). To sum-up, our scheme is better than the one by *Hu (2006)* because of the below reasons.

In terms of the quality of the scheme,

a) our embedding capacity, although lower than *Hu (2006)*, is on the higher side,
b) we obtain higher PSNR values of stego-images as compared to those in *Hu (2006)*,
c) and we further improve the security of the inherently steganographic attack resistant transform based schemes.

In terms of the validation of the scheme,

a) we experiment with a large number of cover images (ten as compared to five in *Hu, 2006*),
b) we check the quality of stego-image on a greater number of numerical measures (five as compared to one in *Hu, 2006*),
c) and we demonstrate the good quality of extracted secret images, which (*Hu, 2006*) do not.

## PROPOSED APPROACH

Our sparse approximation based blind multi-image steganography scheme consists of the following components: (i) Hiding of secret images leading to the generation of the stego-data. (ii) Construction of the stego-image. (iii) Extraction of secret images from the stego-image. These parts are discussed in the respective subsections below.

### Hiding secret images

First, we perform sub-sampling of the cover image to obtain four sub-images. This type of sampling is done because we are hiding up to four secret images. Let $CI$ be the cover image of size $r \times r$. Then, the four sub-images each of size $\frac{r}{2} \times \frac{r}{2}$ are obtained as follows (*Pan et al., 2015*):

$$CI^1(n_1, n_2) = CI(2n_1 - 1, 2n_2 - 1), \tag{1a}$$

$$CI^2(n_1, n_2) = CI(2n_1, 2n_2 - 1), \tag{1b}$$

$$CI^3(n_1, n_2) = CI(2n_1 - 1, 2n_2), \tag{1c}$$

$$CI^4(n_1, n_2) = CI(2n_1, 2n_2), \tag{1d}$$

where $CI^k$, for $k = \{1, 2, 3, 4\}$, are the four sub-images; $n_1, n_2 = 1, 2, \ldots, \frac{r}{2}$ (in our case, $r$ is divisible by 2); and $CI(\cdot, \cdot)$ is the pixel value at $(\cdot, \cdot)$. A cover image and the corresponding four sub-sampled images are shown in Fig. 1.

Originally, these sub-images are not sparse; hence, next, we perform block-wise sparsification of each of these images. For this, we divide each sub-image into blocks of size $b \times b$ and obtain $\frac{r^2}{4 \times b^2}$ blocks for each sub-image (in our case, $b$ divides $r$). Now, we apply discrete cosine transformation to each block. That is,

$$s_i = DCT(x_i), \tag{2}$$

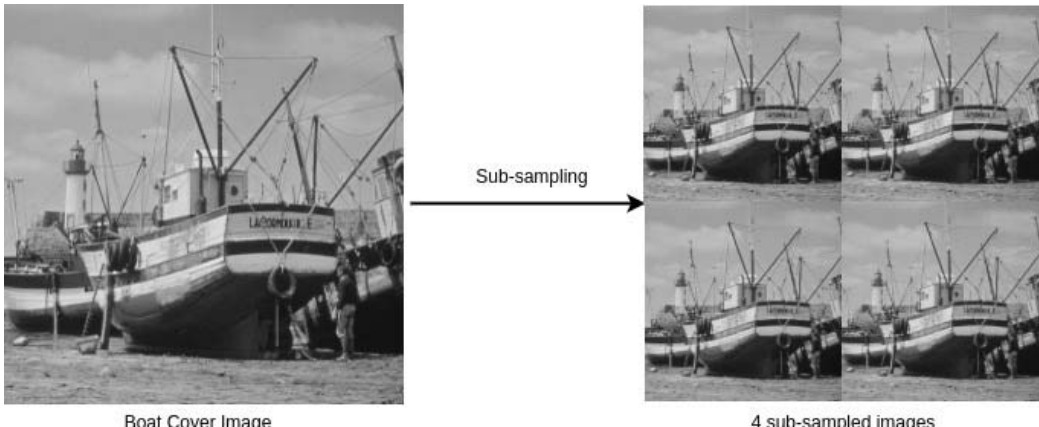

Sub-sampling

Boat Cover Image

4 sub-sampled images

**Figure 1** **A cover image and its 4 sub-sampled images** (*Standard Test Images for Image Prcessing, 2022*). Link: https://github.com/mohammadimtiazz/standard-test-images-for-Image-Processing/blob/master/standard_test_images/boat.png. Copyright: https://github.com/mohammadimtiazz/standard-test-images-for-Image-Processing/blob/master/LICENSE.

where $i = 1, 2, \ldots, \frac{r^2}{4 \times b^2}$, $x_i$ and $s_i$ are the $i$th original and sparse blocks of the same size, i.e, $b \times b$, respectively, and DCT is the Discrete Cosine Transform. Further, we pick the final sparse blocks using a zig-zag scanning order as used in our earlier work (*Pal, Naik & Agrawal, 2019*), and obtain corresponding sparse vectors each of size $b^2 \times 1$. The zig-zag scanning order for a block of size $8 \times 8$ is shown in Fig. 2. This order helps us to arrange the DCT coefficients with the set of large coefficients first, followed by the set of small coefficients, which assists in the preservation of a good quality stego-image.

Next, we represent each vector in two groups based upon large (say #$p_1$) and small (say #$p_2$) coefficients, *i.e.*, $s_{i,u} \in \mathbb{R}^{p_1}$ and $s_{i,v} \in \mathbb{R}^{p_2}$, where $p_1 \leq p_2$. Each of these vectors is sparse and $p_1 + p_2 = b^2$. Further, we oversample each sparse vector using linear measurements as below.

$$y_i = \begin{bmatrix} y_{i,u} \\ y_{i,v} \end{bmatrix} = \begin{bmatrix} s_{i,u} \\ \Phi s_{i,v} \end{bmatrix}, \tag{3}$$

where $y_i \in \mathbb{R}^{(p_1+p_3) \times 1}$ is the set of linear measurements, and $\Phi \in \mathbb{R}^{p_3 \times p_2}$ is the column normalised measurement matrix consisting of normally distributed random numbers with $p_3 > p_2$ and $p_3 \in \mathbb{N}$ (*i.e.*, the sparse coefficients are oversampled).[2] This oversampling helps us to perform sparse approximation. By employing this approximation (along with our novel embedding rule discussed towards the end of this subsection), we achieve a higher embedding capacity. Moreover, our approach gains an extra layer of security because the linear measurements include measurement-matrix encoded small coefficients of the sparse vectors obtained after DCT. Since the distribution of coefficients of the generated sparse vectors is almost the same for all the blocks of an image, we use the same measurement matrix for all the blocks.

Next, we perform processing of the secret images for hiding them into the cover image. Let the size of each secret image be $m \times m$. Initially, we perform block-wise DCT of each of

[2]In the experimental results section, we show how to experimentally pick these coefficients.

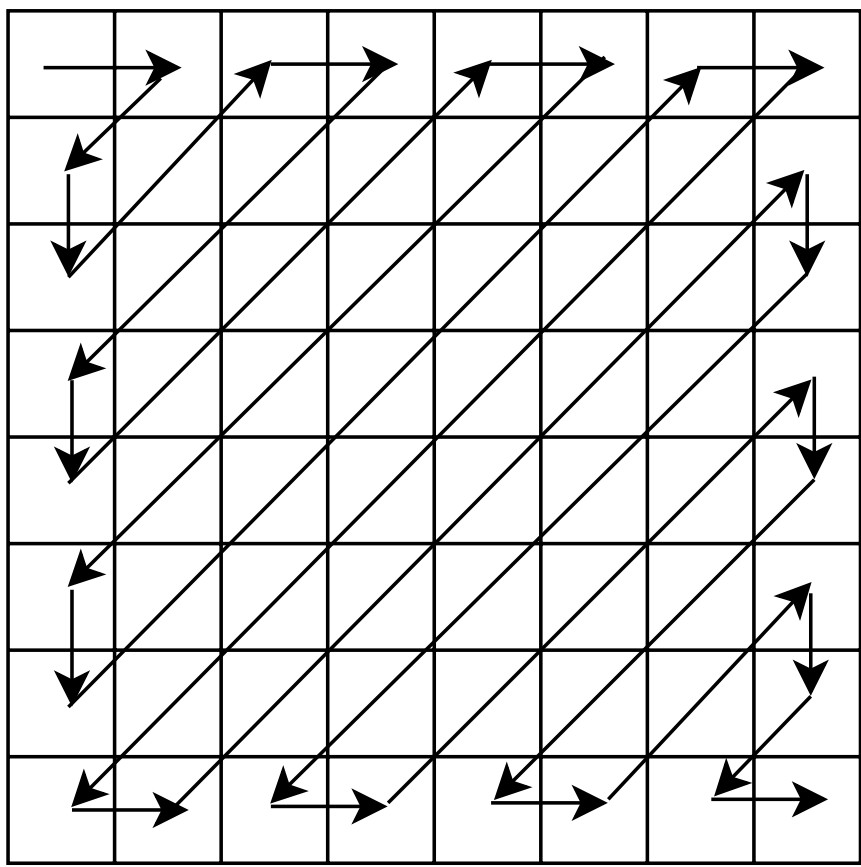

**Figure 2  Zig-zag scanning order for a block of size 8 × 8.**

these images and obtain their corresponding DCT coefficients. Here, the size of each block taken is $l \times l$, and hence, we have $\frac{m^2}{l^2}$ blocks for each secret image. In our case, $l$ divides $m$, and we ensure that $\frac{m^2}{l^2}$ will be less than or equal to $\frac{r^2}{4 \times b^2}$ so that the number of blocks of the secret image is less than or equal to the number of blocks of a cover sub-image. Thereafter, we arrange these DCT coefficients as a vector in the earlier discussed zig-zag scanning order. Let $t_{\hat{i}} \in R^{l^2 \times 1}$, for $\hat{i} = 1, 2, \ldots, \frac{m^2}{l^2}$, be the vector representation of the DCT coefficients of one secret image. We pick the initial $p_4$ DCT coefficients with relatively larger values (out of the available $l^2$ coefficients) for hiding,[3] where $p_4 \in \mathbb{N}$. Omitting the remaining coefficients $(l^2 - p_4)$ does not significantly deteriorate the quality of the extracted secret image.

Here, we show the hiding of only one secret image into one sub-image of the cover image. However, in our steganography scheme, we can hide a maximum of four secret images, one in each of the four sub-images of the cover image, which is demonstrated in the experimental results section. If we want to hide less than four secret images, we can randomly select the corresponding sub-images from the available four.

Next, using our novel embedding rule (discussed below), we hide the chosen $p_4$ DCT coefficients of the secret image into a selected set of $p_1 + p_3$ linear measurements obtained

[3]In the experimental results section, we show how to experimentally pick these coefficients.

**Table 6** The detail of hiding secret image coefficients into the linear measurement coefficients of the cover image.

| | Secret image coefficient indices | |
|---|---|---|
| 1 | 2 to c | $c+1$ to $p_4$ |
| | Companion linear measurement coefficient indices | |
| $p_1 - 2c$ | $p_1 - 2c + 1$ to $p_1 - c - 1$ | $p_1 + c + 1$ to $p_1 + p_4$ |
| | Replaced linear measurement coefficient indices | |
| $p_1$ | $p_1 - c + 1$ to $p_1 - 1$ | $p_1 + p_4 + 1$ to $p_1 + 2 \times p_4 - c$ |

from the sub-image of the cover image, leading to the generation of the stego-data (we ensure that $p_4$ is less than $p_1 + p_3$).

We hide secret image data into the cover image by taking linear combinations of each secret image coefficient with a companion linear measurement coefficient of the cover image. These linear combinations replace certain other linear coefficients of the cover image to obtain the so called stego-data (subsequently, stego-image). The three groups of index coefficients are listed in Table 6.

The data in Table 6 is based upon three design choices as below.

a) As can be seen from Table 6, we divide each group of coefficients into three ranges in a staggered manner to achieve a higher level of security.

b) The specific choice of indices in the second and fourth rows of Table 6 is made so as to hide secret image coefficients in relatively small valued cover image coefficients (companion linear measurement coefficients). This results in a relatively improved quality stego-image.

c) In Table 6, the replaced linear measurement coefficient indices differ just slightly from the chosen companion coefficient indices (fourth and sixth rows respectively). The reason for this is that we want our extraction rule (discussed in section 'Extraction of the secret images') to be as less lossy as possible, resulting in less deteriorated extracted secret images.

The whole process is given in **Algorithm** 1. Specifically, the indices discussed in Table 6 are given on line 3, lines 4–6, and lines 7–9 of this algorithm, respectively. The block diagram for this complete data embedding process is given in Fig. 3. A small numerical example, which further explains this hiding process is given in Appendix, 'A small numerical example of our embedding process'.

## Construction of the Stego-Image

As mentioned earlier, the next step in our scheme is the construction of the stego-image. Since we can hide a maximum of four secret images into four sub-images of a single cover image, we first construct four sub-stego-images and then perform inverse sampling to obtain a single stego-image. Let $s_i'$ be the sparse vector of the $i$th block of a sub-stego-image. The sparse vector $s_i'$ is the concatenation of $s_{i,u}'$ and $s_{i,v}'$. Here, the size of $s_{i,u}'$, $s_{i,v}'$, and $s'$ is the same as that of $s_{i,u}$, $s_{i,v}$, and $s$, respectively. Then, we have

$$s_{i,u}' = y_{i,u}', \tag{4a}$$

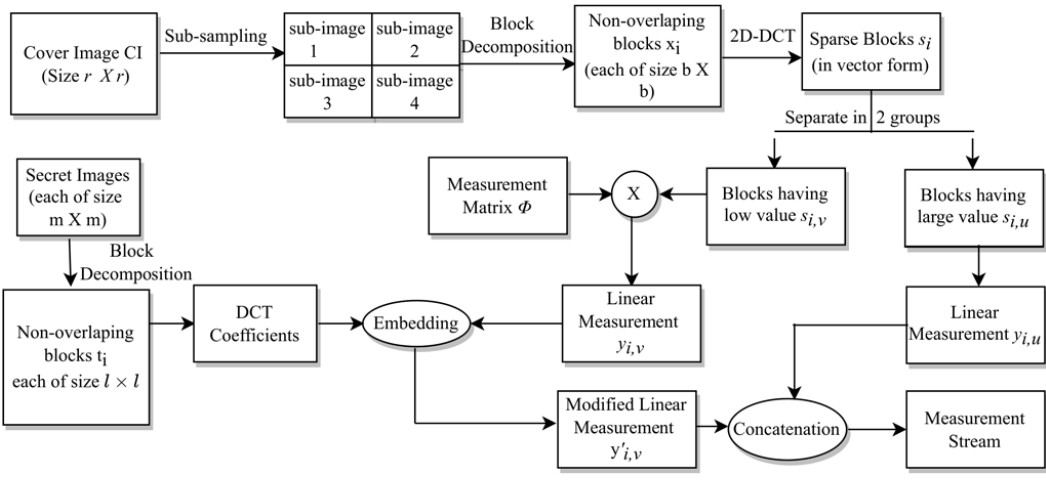

**Figure 3** The embedding process.

$$s'_{i,v} = \underset{s'_{i,v} \in \mathbb{R}^{p2}}{\arg\min} \|s'_{i,v}\|_1 \quad \text{subject to} \quad \Phi s'_{i,v} = y'_{i,v}, \tag{4b}$$

where $y'_i$ is defined in **Algorithm** 1, and it is equal to $\begin{bmatrix} y'_{i,u} \\ y'_{i,v} \end{bmatrix}$ as split in Eq. (3). The second part Eq. (4b) (*i.e.*, obtaining $s_{i,v}'$), is an $\ell_1$-norm minimization problem. Here, we can observe that in the above optimization problem, the constraints are oversampled. As earlier, this oversampling helps us to do sparsification, which leads to increased embedding capacity as well as increased security because the measurement matrix is encoded. For the solution of the minimization problem Eq. (4b), we use ADMM (*Boyd et al., 2010*; *Gabay, 1976*) to solve the LASSO (*Hwang, Kim & Kim, 2016*; *Nardone, Ciaramella & Staiano, 2019*) formulation of this minimization problem.[4] We use this method because it has a fast convergence, is easy to implement, and also is extensively used in image processing (*Boyd et al., 2010*; *Hwang, Kim & Kim, 2016*).

Next, we convert each vector $s_i'$ into a block of size $b \times b$. After that, we apply inverse discrete cosine transformation (*i.e.*, the two-dimensional Inverse DCT) to each of these blocks to generate blocks $x_i'$ of the image. That is,

$$x_i' = IDCT\left(s_i'\right). \tag{5}$$

Next, we construct the sub-stego-image of size $\frac{r}{2} \times \frac{r}{2}$ by arranging all these blocks $x_i'$. We repeat the above steps to construct all four sub-stego-images. At last, we perform inverse sampling and obtain a single constructed stego-image from these four sub-stego-images. In the experiments section, we show that the quality of the stego-image is also very good. The block representation of these steps is given in Fig. 4. A small numerical example, which further explains this process is given in Appendix, 'A small numerical example of our embedding process'.

[4]Since the linear system of equations in (4b) is overdetermined, we solve it in least squares sense that causes loss of information.

---

**Algorithm 1** Embedding Rule

**Input:**

- $y_i$: Sequence of linear measurements of the cover image with $i = 1, 2, \ldots, \frac{r^2}{4 \times b^2}$.
- $t_{\hat{i}}$: Sequence of transform coefficients of the secret image with $\hat{i} = 1, 2, \ldots, \frac{m^2}{l^2}$.
- The choice of our $r$, $b$, $m$, and $l$ is such that $\frac{m^2}{l^2}$ is less than or equal to $\frac{r^2}{4 \times b^2}$.
- $p_1$ and $p_4$ are lengths of certain vectors defined on pages ix and x, respectively.
- $\alpha$, $\beta$, $\gamma$, and $c$ are algorithmic constants that are chosen based upon experience. The choices of these constants are discussed in the experimental results sections.

**Output:**

- $y_i'$: The modified version of the linear measurements with $i = 1, 2, \ldots, \frac{r^2}{4 \times b^2}$.

1:  Initialize $y_i'$ to $y_i$, where $i = 1, 2, \ldots, \frac{r^2}{4 \times b^2}$.
2:  **for** $\hat{i} = 1$ to $\frac{m^2}{l^2}$ **do**
3:      // Embedding of the first coefficient.
        $$y_i'(p_1) = y_i(p_1 - 2c) + \alpha \times t_i(1).$$
4:      **for** $j = p_1 - c + 1$ to $p_1 - 1$ **do**
5:          // Embedding of the next $c - 1$ coefficients.
            $$y_i'(j) = y_i(j - c) + \beta \times t_i(j - p_1 + c + 1).$$
6:      **end for**
7:      **for** $k = p_1 + p_4 + 1$ to $p_1 + 2 \times p_4 - c$ **do**
8:          // Embedding of the remaining $p_4 - c$ coefficients.
            $$y_i'(k) = y_i(k - p_4 + c) + \gamma \times t_i(k - p_1 - p_4 + c).$$
9:      **end for**
10: **end for**
11: **return** $y_i'$

---

## Extraction of the secret images

In this subsection, we discuss the process of extracting secret images from the stego-image. Initially, we perform sampling (as done in section 'Hiding Secret Images' using Eq. (1a)–Eq. (1d)) of the stego-image to obtain four sub-stego-images. Since the extraction of all the secret images is similar, here, we discuss the extraction of only one secret image from one sub-stego-image. First, we perform block-wise sparsification of the chosen sub-stego-image. For this, we divide the sub-stego-image into blocks of size $b \times b$. We obtain a total of $\frac{r^2}{4 \times b^2}$ blocks. Further, we sparsify each block (say $x_i'$) by computing the corresponding sparse vector (say $s_i'$). That is,

$$s_i' = DCT(x_i'). \tag{6}$$

Next, as earlier, we arrange these sparse blocks in a zig-zag scanning order, obtain the corresponding sparse vectors each of size $b^2 \times 1$, and then categorize each of them into two

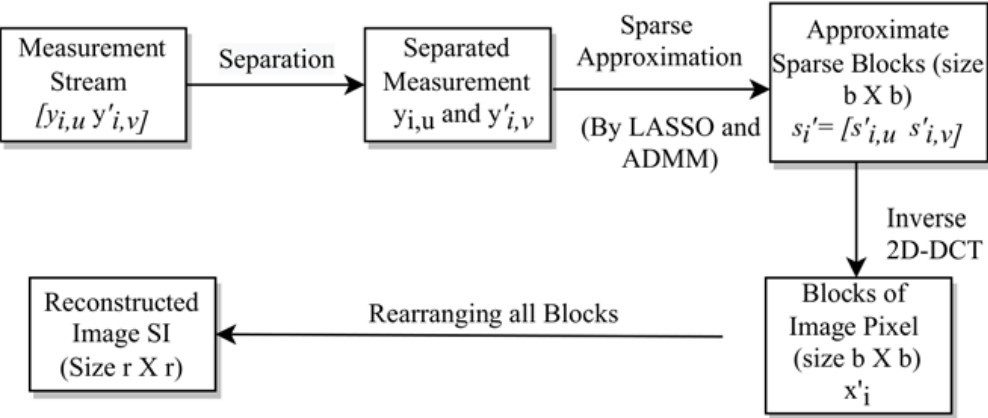

**Figure 4** Stego-image construction.

groups $s'_{i,u} \in \mathbb{R}^{p_1}$ and $s'_{i,v} \in \mathbb{R}^{p_2}$. Here, as before, $p_1$ and $p_2$ are the numbers of coefficients having large values and small values (or zero values), respectively. After that, we oversample each sparse vector using linear measurements (say $y'_i \in \mathbb{R}^{(p_1+p_3)\times 1}$),

$$y'_i = \begin{bmatrix} y'_{i,u} \\ y'_{i,v} \end{bmatrix} = \begin{bmatrix} s'_{i,u} \\ \Phi s'_{i,v} \end{bmatrix}. \qquad (7)$$

From $y'_i$, we extract the DCT coefficients of the embedded secret image using **Algorithm 2**. This extraction rule is the reverse of the embedding rule given in **Algorithm 1**.

In **Algorithm 2**, $t'_{\hat{i}} \in \mathbb{R}^{l^2 \times 1}$, for $\hat{i} = 1, 2, \ldots, \frac{m^2}{l^2}$, are the vector representations of the DCT coefficients of the blocks of one extracted secret image. Next, we convert each vector $t'_{\hat{i}}$ into blocks of size $l \times l$, and then perform a block-wise Inverse DCT (IDCT) (using Eq. (5)) to obtain the secret image pixels. Finally, we obtain the extracted secret image of size $m \times m$ by arranging all these blocks column wise. As mentioned earlier, this steganography scheme is a blind multi-image steganography scheme because it does not require any cover image data at the receiver side for the extraction of secret images.

Here, the process of hiding (and extracting) secret images is not fully lossless,[5] resulting in the degradation of the quality of extracted secret images. This is because we first oversample the original image using Eq. (3), and then we construct the stego-image by solving the optimization problem (4b), which leads to a loss of information. However, our algorithm is designed in such a way that we are able to extract high-quality secret images. We support this fact with examples in the experiments section (specifically, 'Secret Image Quality Assessment'). We term our algorithm Sparse Approximation Blind Multi-Image Steganography (SABMIS) scheme due to the involved sparse approximation and the blind multi-image steganography.

The above extraction process is represented *via* a block diagram in Fig. 5. As discussed earlier, this extraction is just the reverse of the embedding process.

[5]This is common in transform-based image steganography.

**Algorithm 2** Extraction Rule

**Input:**

- $y_i''$: Sequence of linear measurements of the stego-image with $i = 1, 2, \ldots, \frac{r^2}{4 \times b^2}$.
- $p_1, p_4, \alpha, \beta, \gamma$, and $c$ are chosen as in **Algorithm 1**.

**Output:**

- $t_{\hat{i}}'$: Sequence of transform coefficients of the secret image with $\hat{i} = 1, 2, \ldots, \frac{m^2}{l^2}$.

1:  Initialize $t_{\hat{i}}'$ to zeros, where $\hat{i} = 1, 2, \ldots, \frac{m^2}{l^2}$.

2:  **for** $\hat{i} = 1$ to $\frac{m^2}{l^2}$ **do**

3:      // Extraction of the first coefficient.
$$t'_{\hat{i}}(1) = \frac{y_i''(p_1) - y_i''(p_1 - 2c)}{\alpha}.$$

4:      **for** $j = p_1 - c + 1$ to $p_1 - 1$ **do**

5:          // Extraction of the next $c - 1$ coefficients.
$$t_{\hat{i}}'(j - p_1 + c + 1) = \frac{y_i''(j) - y_i''(j - c)}{\beta}.$$

6:      **end for**

7:      **for** $k = p_1 + p_4 + 1$ to $p_1 + 2 \times p_4 - c$ **do**

8:          // Extraction of the remaining $p_4 - c$ coefficients.
$$t_{\hat{i}}'(k - p_1 - p_4 + c) = \frac{y_i''(k) - y_i''(k - p_4 + c)}{\gamma}.$$

9:      **end for**

10: **end for**

11: **return** $t_{\hat{i}}'$

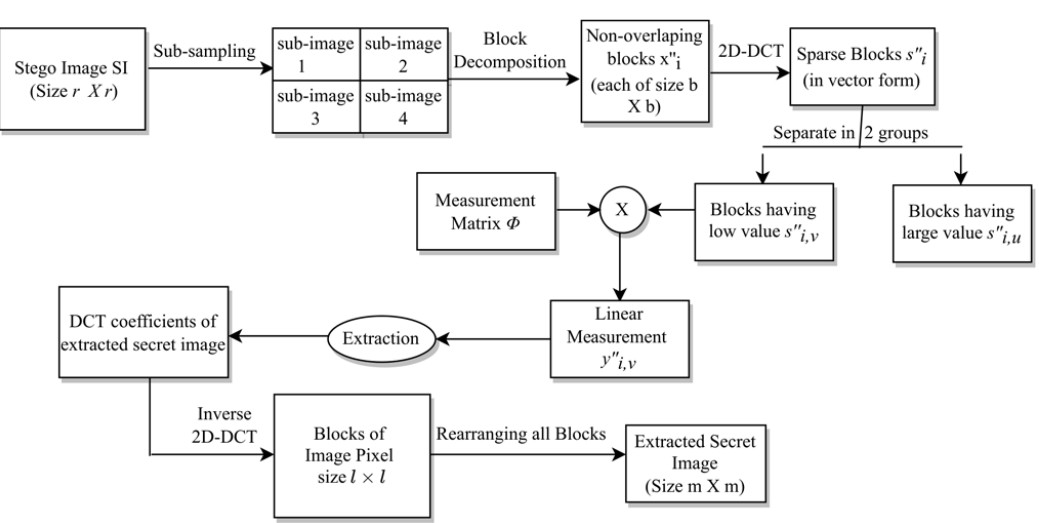

**Figure 5** The extraction process.

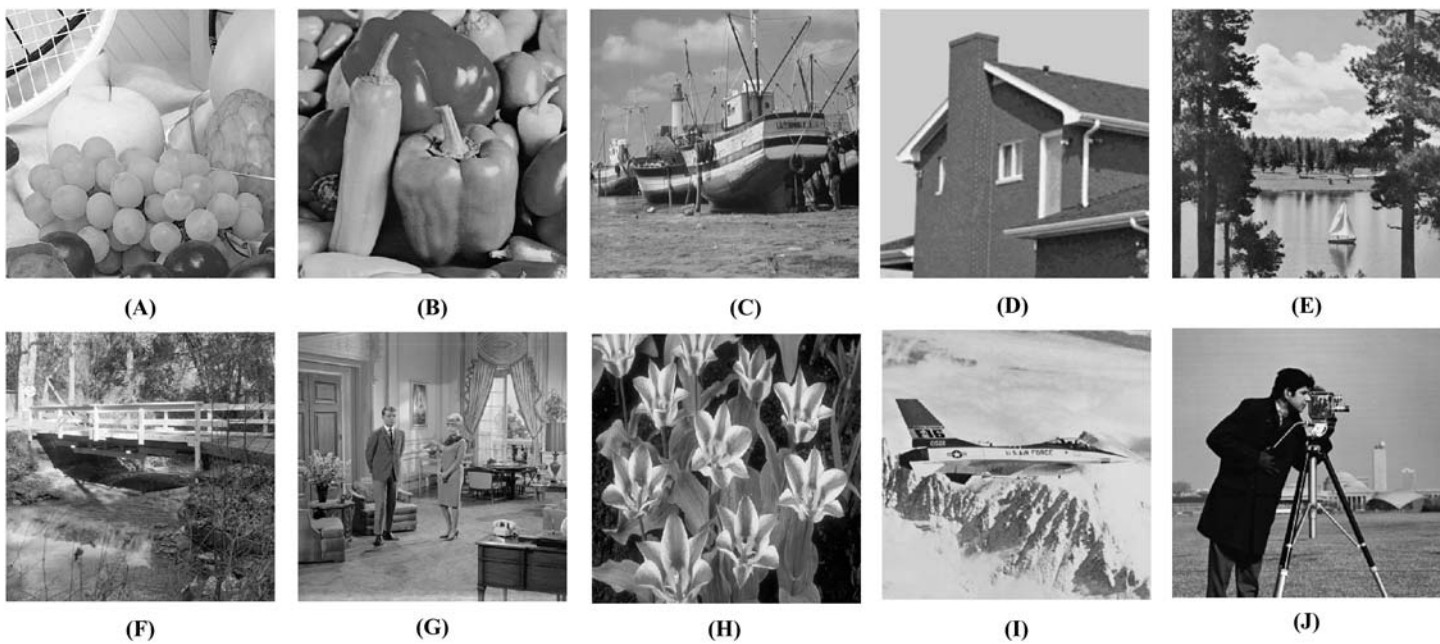

**Figure 6** **Test images used in our experiments (*Standard Test Images for Image Prcessing, 2022*).** (A) Fruits, (B) Peppers, (C) Boat, (D) House, (E) Lake, (F) Stream, (G) Living room, (H) Tulips, (I) Airplane, and (J) Camera man. Links: (A) https://github.com/mohammadimtiazz/standard-test-images-for-Image-Processing/blob/master/standard_test_images/fruits.png (B) https://raw.githubusercontent.com/mohammadimtiazz/standard-test-images-for-Image-Processing/master/standard_test_images/peppers_gray.tif (C) https://github.com/mohammadimtiazz/standard-test-images-for-Image-Processing/blob/master/standard_test_images/boat.png (D) https://raw.githubusercontent.com/mohammadimtiazz/standard-test-images-for-Image-Processing/master/standard_test_images/house.tif (E) https://raw.githubusercontent.com/mohammadimtiazz/standard-test-images-for-Image-Processing/master/standard_test_images/lake.tif (F) https://raw.githubusercontent.com/mohammadimtiazz/standard-test-images-for-Image-Processing/master/standard_test_images/walkbridge.tif (G) https://raw.githubusercontent.com/mohammadimtiazz/standard-test-images-for-Image-Processing/master/standard_test_images/livingroom.tif (H) https://github.com/mohammadimtiazz/standard-test-images-for-Image-Processing/blob/master/standard_test_images/tulips.png (I) https://raw.githubusercontent.com/mohammadimtiazz/standard-test-images-for-Image-Processing/master/standard_test_images/jetplane.tif (J)https://raw.githubusercontent.com/mohammadimtiazz/standard-test-images-for-Image-Processing/master/standard_test_images/cameraman.tif. Copyright: https://github.com/mohammadimtiazz/standard-test-images-for-Image-Processing/blob/master/LICENSE.

## EXPERIMENTAL RESULTS

Experiments are carried out in MATLAB on a machine with an Intel Core i5 processor @2.50 GHz and 8GB RAM. We use 10 standard test images (those which are frequently found in the literature) for our experiments. These image are freely available with no copyright (*Standard Test Images for Image Prcessing, 2022*).

Here, we take all ten images shown in Fig. 6 as the cover images, and four images; Figs. 6B, 6E, 6F, and 6J as the secret images for our experiments. However, we can use any of the ten images as the secret images.

Although the images shown in Fig. 6 look to be of the same dimension, they are of varying sizes. For our experiments, each cover image is converted to $1024 \times 1024$ size (*i.e., $r \times r$*). We take blocks of size $8 \times 8$ for the cover images (*i.e., $b \times b$*). Recall from section 'Hiding Secret Images' that the size of the DCT sparsified vectors is $(p_1 + p_2) \times 1$ with $p_1 + p_2 = b^2$ (here, $b^2 = 64$). In general, applying DCT on images results in sparse vectors where more

than half of the coefficients have values that are either very small or zero (*Agrawal & Ahuja, 2021*; *Pal, Naik & Agrawal, 2019*; *Pan et al., 2015*). This is the case here as well. Hence, in our experiments, we take $p_1 = p_2 = 32$. Recall, the size of the measurement matrix $\Phi$ is $p_3 \times p_2$ with $p_3 > p_2$. We take $p_3 = 50 \times p_2$. Without loss of generality, the element values of the column-normalized measurement matrix are taken as random numbers with mean 0 and standard deviation 1, which is a common standard.

There are many options for taking the size of the secret images. In one way the size of the length and the width of the secret image is taken to be the same as the length and the width of the cover image (*Sanjutha, 2018*). In another approach, which many papers follow, the dimensions of the secret image are taken to be substantially smaller than the dimensions of the cover image. For example, the size of the length and the width of the secret image to be half of the length and the width of the cover image (*Hemalatha et al., 2013*; *Arunkumar et al., 2019a*; *Arunkumar et al., 2019b*), respectively. Another option is to use a factor of one-fourth (*Manujala & Danti, 2015*). Hence, without any loss of generality, we take the dimensions of the secret image to be half of the dimensions of the cover image.

Thus, each secret image is converted to $512 \times 512$ size (*i.e.*, $m \times m$). We take blocks of size $8 \times 8$ for the secret images as well (*i.e.*, $l \times l$). In general, the DCT coefficients can be divided into three sets (*Shastri, Tamrakar & Ahuja, 2018*); low frequencies, middle frequencies, and high frequencies. Low frequencies are associated with the illumination, middle frequencies are associated with the structure, and high frequencies are associated with the noise or small variation details. Thus, these high-frequency coefficients are of very little importance for the to-be embedded secret images. Since the number of high-frequency coefficients is usually half of the total number of coefficients, we take $p_4 = 32$ (using $8 \times 8$ divided by 2).

The values of the constants in **Algorithm 1** and **Algorithm 2** are taken as follows[6] (based upon experience): $\alpha = 0.01$, $\beta = 0.1$, $\gamma = 1$, and $c = 6$. The LASSO constant is taken as $\lambda = 0.011\lambda_{max}$, where $\lambda_{max} = \|\Phi^T y'_{i,v}\|_\infty$ with $\| \cdot \|_\infty$ being the $\ell_\infty$-norm (*Agrawal et al., 2021*). For ADMM, we set the absolute stopping tolerance as $1 \times 10^{-4}$, and the relative stopping tolerance as $1 \times 10^{-2}$. These values are again taken based upon our experience with a similar algorithm (*Agrawal et al., 2021*). Eventually, our ADMM always converges in 5 to 20 iterations.

As mentioned earlier, in the six sections below we experimentally demonstrate the usefulness of our steganography scheme. In section 'Embedding Capacity Analysis', we show analytically that our SABMIS scheme gives excellent embedding capacities. In section 'Stego-Image Quality Assessment', we show that the quality of the constructed stego-images, when compared with the corresponding cover images, is high. In section 'Secret Image Quality Assessment', we demonstrate the good quality of the extracted secret images when compared with the original secret images. In section 'Security Analysis', we show that our SABMIS scheme is resistant to steganographic attacks. In section 'Timing Data', we demonstrate efficiency of SABMIS by providing its timing data. In section 'Application of Our Scheme on Real-life Data', we discuss applicability of our scheme to real-life data, and hence, demonstrate its practical usefulness.

[6] The values of these constants do not affect the convergence of ADMM much. Determining the range of values that work best here is part of our future work.

## Embedding capacity analysis

The embedding capacity (or embedding rate) is the number (or length) of secret bits that can be hidden/ embedded in each pixel of the cover image. It is measured in bits per pixel[7] (bpp) and is calculated as follows:

$$\text{EC in bpp} = \frac{\text{Total number of secret bits embedded}}{\text{Total number of pixels in the cover image}}. \tag{8}$$

As motivated on the previous page, we chose the size of the length and the width of secret image to be half of the length and the width of cover image, respectively. Since our cover images are of size $1024 \times 1024$, our secret images are taken to be of size $512 \times 512$. For a grayscale image, each pixel size is 8 bits. Hence, when hiding one secret image in a cover image, we obtain the embedding capacity as below.

$$\text{EC in bpp} = \frac{512 \times 512 \times 8}{1024 \times 1024}, \tag{9}$$

which is equal to 2 bpp. Similarly, while hiding two, three, and four secret images in a cover image, we obtain the embedding capacities of 4 bpp, 6 bpp, and 8 bpp, respectively.

## Stego-image quality assessment

In general, the visual quality of the stego-image degrades as the embedding capacity increases. Hence, preserving the visual quality becomes increasingly important. There is no universal criterion to determine the quality of the constructed stego-image. However, we evaluate it by visual and numerical measures. We use Peak Signal-to-Noise Ratio (PSNR), Mean Structural Similarity (MSSIM) index, Normalized Cross-Correlation (NCC) coefficient, entropy, and Normalized Absolute Error (NAE) numerical measures.

When using the visual measures, we construct the stego-images corresponding to the different cover images used in our experiments and then check their distortion visually. We also check their corresponding edge map diagrams. Here, we present the visual comparison only for 'Stream' as the cover image with 'Lake' secret image and the corresponding stego-image. We get similar results for the other images as well. The comparison is given in Fig. 7. The cover image and its corresponding edge map are shown in parts (A) and (B) of this figure. The stego-image and its corresponding edge map are given in parts (C) and (D) of the same figure. When we compare each figure with its counterpart, we find that they are very similar.

Next, when using the numerical measures to assess the quality of the stego-image with respect to the cover image, we first evaluate the most common measure of PSNR value in section 'Peak Signal-to-Noise Ratio (PSNR) Value'. Subsequently, we evaluate the other more rarely used numerical measures of MSSIM index, NCC coefficient, entropy, and NAE in section 'Other Numerical Measures'.

### Peak Signal-to-Noise Ratio (PSNR) value

We compute the *PSNR* values to evaluate the imperceptibility of stego-images (SI) with respect to the corresponding cover images (CI) as follows (*Elzeki et al., 2021*):

$$PSNR(CI, SI) = 10\log_{10}\frac{R^2}{MSE(CI, SI)}\ dB, \tag{10}$$

[7] Since in the transform domain-based steganography schemes, some specific transform coefficients are hidden into the cover image (along with the secret bits), a more appropriate term that can be used for embedding capacity is "bits of information per pixel" (bipp). However, to avoid confusion, we use the term bpp in this article, which is commonly used.

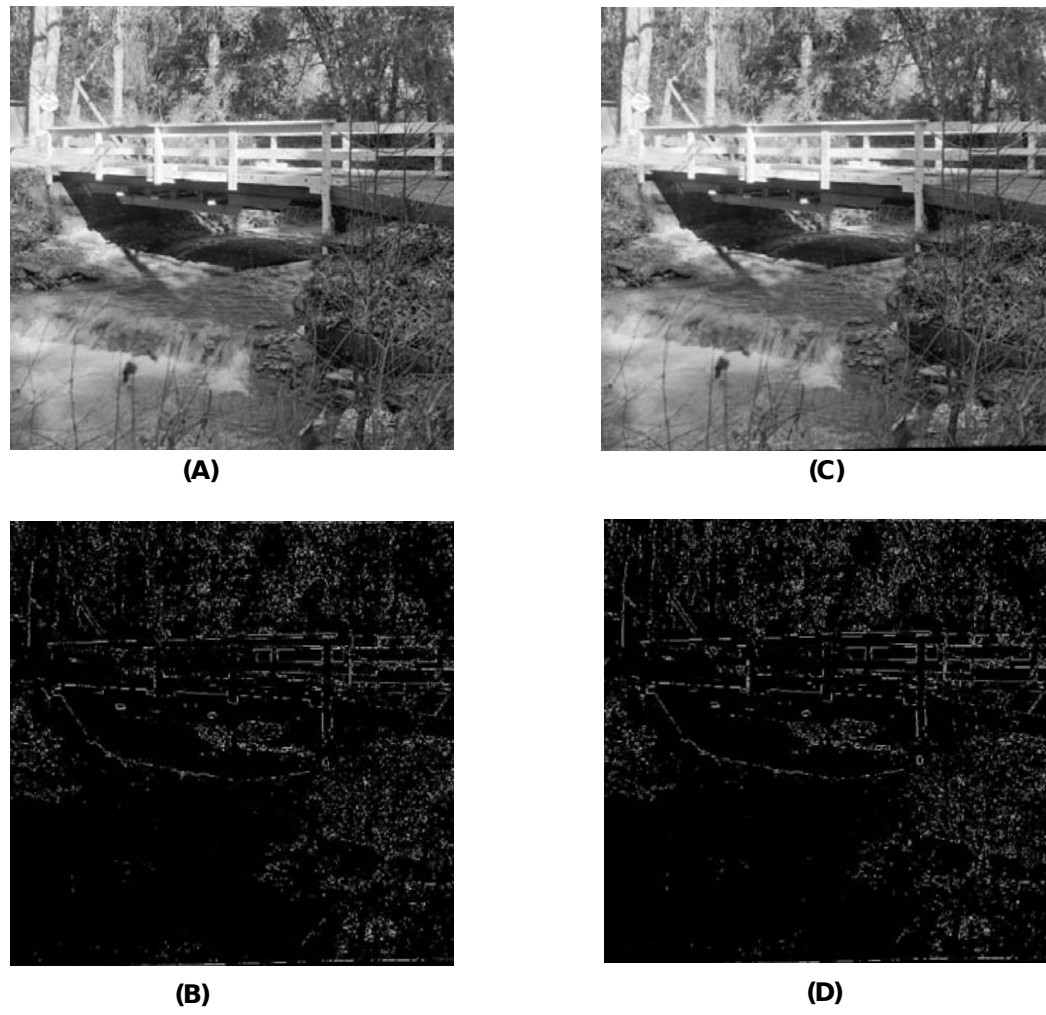

**Figure 7** **Visual quality analysis between 'Stream' cover image (CI) and its corresponding stego-image (SI) (with 'Lake' secret image embedded in it).** (A) Cover image, (B) cover image edge map, (C) stego-image, and (D) stego-image edge map. Link: https://raw.githubusercontent.com/mohammadimtiazz/ standard-test-images-for-Image-Processing/master/standard_test_images/walkbridge.tif. Copyright: https://github.com/mohammadimtiazz/standard-test-images-for-Image-Processing/blob/master/LICENSE.

where $R$ is the maximum intensity of the pixels, which is 255 for grayscale images, dB refers to decibel, and $MSE(CI, SI)$ represents the mean square error between the cover image $CI$ and the stego-image $SI$ that is calculated as

$$MSE(CI, SI) = \frac{\sum_{i=1}^{r1} \sum_{j=1}^{r2} \left( CI(i,j) - SI(i,j) \right)^2}{r1 \times r2}, \tag{11}$$

where $r1$ and $r2$ represent the row and column numbers of the image (for us either cover or stego), respectively, and $CI(i,j)$ and $SI(i,j)$ represent the pixel values of the cover image and the stego-image, respectively.

A higher PSNR value indicates a higher imperceptibility of the stego-image with respect to the corresponding cover image. In general, a value higher than 30 dB is considered to

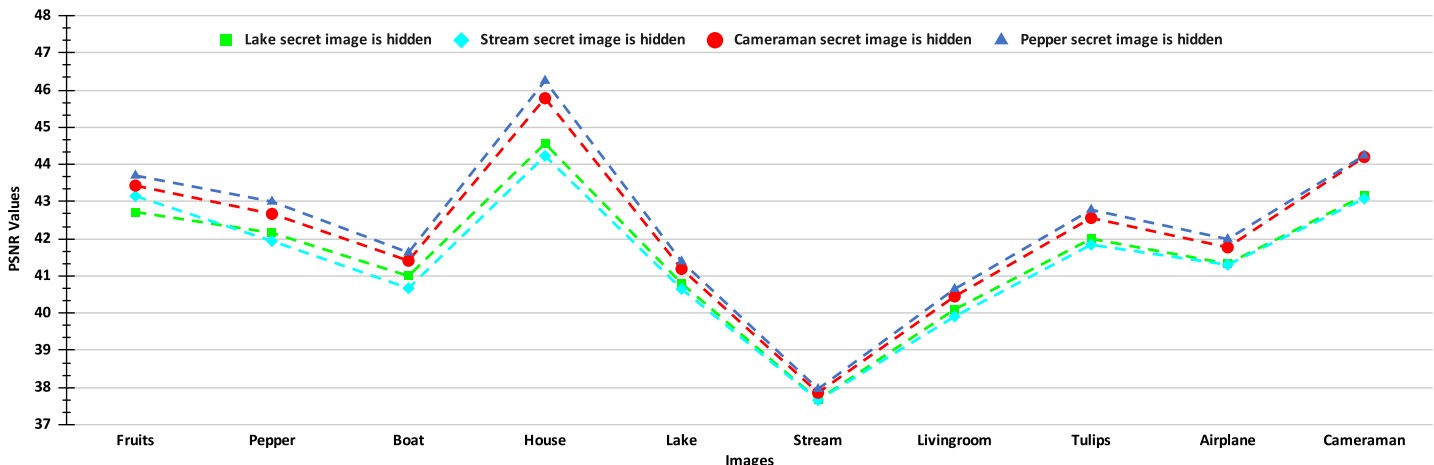

**Figure 8** PSNR values of the stego-images when only one secret image is hidden.

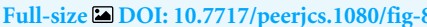

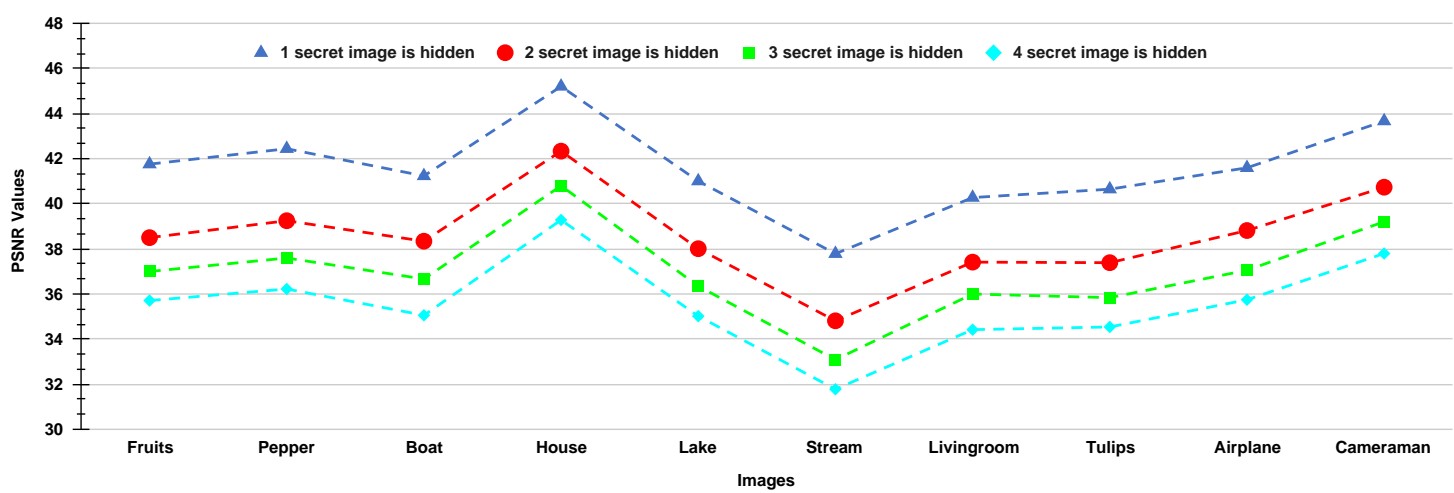

**Figure 9** PSNR values of the stego-images when different numbers of images are hidden.

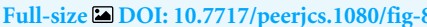

be good since human eyes can hardly distinguish the distortion in the image (*Gutub & Shaarani, 2020*; *Zhang et al., 2013*; *Liu & Liao, 2008*).

The PSNR values of the stego-images corresponding to the ten cover images are given in Figs. 8 and 9. In Fig. 8, we show the PSNR values of all the stego-images when separately all the four secret images (mentioned above in Fig. 6) are hidden. In this figure, we obtain the highest PSNR value (46.25 dB) when the 'Peppers' secret image is hidden in the 'House' cover image, while the lowest PSNR value (37.66 dB) is obtained when the 'Stream' secret image is hidden in the 'Stream' cover image.

In Fig. 9, we show the PSNR values for the four cases of hiding one, two, three, and four secret images in the ten cover images. As we have four secret images, when hiding one secret image, we have a choice of hiding any one of them and present the resulting PSNR

values. However, we separately hide all four images, obtain their PSNR values, and then present the average results. Similarly, the average PSNR values are presented for the cases when we hide two and three images. In this figure, we obtain the highest average PSNR value (45.21 dB) when one secret image is hidden in the 'House' cover image, while the lowest PSNR value (31.78 dB) is obtained when all four secret images are hidden in the 'Stream' cover image. Also, we observe that for all test cases, we obtain PSNR values higher than 30 dB which, as earlier, are considered good.

### Other numerical measures

**Mean Structural Similarity (MSSIM) Index**: This is an image quality assessment metric used to measure the structural similarity between two images, which is most noticeable to humans (*Habib et al., 2016*; *Elzeki et al., 2021*). MSSIM between the cover image $CI$ and the stego-image $SI$ is given as

$$MSSIM(CI, SI) = \frac{1}{M} \sum_{j=1}^{M} SSIM(ci_j, si_j), \tag{12}$$

where $ci_j$ and $si_j$ are the pixel values of the cover image and the stego-image, respectively, at the $j$th local window[8] with $M$ being the number of local windows (*Habib et al., 2016*; *Wang et al., 2004*), and

$$SSIM(x, y) = \frac{(2\mu_x \mu_y + C_1)(2\sigma_{xy} + C_2)}{(\mu_x^2 + \mu_y^2 + C_1)(\sigma_x^2 + \sigma_y^2 + C_2)}, \tag{13}$$

where for vectors $x$ and $y$; $\mu_x$ is the weighted mean of $x$; $\mu_y$ is the weighted mean of $y$; $\sigma_x$ is the weighted standard deviation of $x$; $\sigma_y$ is the weighted standard deviation of $y$; $\sigma_{xy}$ is the weighted covariance between $x$ and $y$; $C_1$ and $C_2$ are positive constants.

We take $M = 1069156$, $C_1 = (0.01 \times 255)^2$, and $C_2 = (0.03 \times 255)^2$ based upon the recommendations from *Habib et al. (2016)*; *Wang et al. (2004)*. The value of the MSSIM index lies between 0 and 1, where the value 0 indicates that there is no structural similarity between the cover image and the corresponding stego-image, and the value 1 indicates that the images are identical.

**Normalized Cross-Correlation (NCC) Coefficient:** This metric measures the amount of correlation between two images (*Parah et al., 2016*). The NCC coefficient between the cover image $CI$ and the stego-image $SI$ is given as

$$NCC(CI, SI) = \frac{\sum_{i=1}^{r1} \sum_{j=1}^{r2} CI(i,j)SI(i,j)}{\sum_{i=1}^{r1} \sum_{j=1}^{r2} CI^2(i,j)}, \tag{14}$$

where $r1$ and $r2$ represent the row and column numbers of the image (for us either cover or stego), respectively, and $CI(i,j)$ and $SI(i,j)$ represent the pixel values of the cover image and the stego-image, respectively. The NCC coefficient value of 0 indicates that the cover image and the stego-image are not correlated while a value of 1 indicates that the two are highly correlated.

**Entropy:** In general, entropy is defined as the measure of average uncertainty of a random variable. In the context of an image, it is a statistical measure of randomness that can

[8] It is a 11×11 Gaussian matrix, which is standard in the calculation of MSSIM.

be used to characterize the texture of the image (*Gonzalez, Woods & Eddins, 2004*). For a grayscale image (either a cover image or a stego-image in our case), entropy is given as

$$E = -\sum_{i=0}^{255} (p_i \log_2 p_i), \tag{15}$$

where $p_i \in [0, 1]$ is the fraction of image pixels that have the value $i$. If the stego-image is similar to its corresponding cover image, then the two should have similar entropy values (due to similar textures).

**Normalized Absolute Error (NAE):** This metric is a distance measure that captures pixel-wise differences between two images (*Arunkumar et al., 2019b*). NAE between the cover image *CI* and the stego-image *SI* is given as

$$NAE(CI, SI) = \frac{\sum_{i=1}^{r1} \sum_{j=1}^{r2} \left( |CI(i,j) - SI(i,j)| \right)}{\sum_{i=1}^{r1} \sum_{j=1}^{r2} CI(i,j)}, \tag{16}$$

where $r1$ and $r2$ represent the row and the column numbers of the image (for us either cover or stego), respectively, and $CI(i,j)$ and $SI(i,j)$ represent the pixel values of the cover image and the stego-image, respectively. NAE has values in the range 0 to 1. A value close to 0 indicates that the cover image is very close to its corresponding stego-image, and a value close to 1 indicates that the two are substantially far apart.

In Table 7, we present the values of MSSIM index, NCC coefficient, entropy and NAE for our SABMIS scheme when hiding all four secret images. We do not present the values for the cases of embedding less than four secret images as their results will be better than those given in Table 7. Hence, our reported results are for the worst case. From this table, we observe that all values of the MSSIM index are nearly equal to 1 (different in the sixth place of decimal), the values of NCC coefficients are close to 1, and values of NAE are close to 0. The entropy values of the cover and the stego-images are almost identical. All these values indicate that the cover images and their corresponding stego-images are almost identical.

### Secret image quality assessment

Since human observers are considered the final arbiter to assess the quality of the extracted secret images, we compare one such original secret image and its corresponding extracted secret image. The results of all other combinations are almost the same. In Figs. 10A and 10C, we show the original 'Lake' secret image and the extracted 'Lake' secret image (from the 'Stream' stego-image). From these figures, we observe that there is little distortion in the extracted image. Besides this, for these two images, we also present their corresponding edge map diagrams (in Figs. 10B and 10D, respectively). Again, we observe minimal variations between the original and the extracted secret images.

### Security analysis

The SABMIS scheme is a transform domain based technique which employs an indirect embedding strategy, *i.e.*, it does not follow the Least Significant Bits (LSB) flipping method, and hence, it is immune to statistical attacks (*Westfeld & Pfitzmann, 2000*; *Yu et al., 2009*).

**Table 7** MSSIM index, NCC coefficient, entropy, and NAE of the stego-images when compared with the corresponding cover images.

| Cover image | MSSIM | NCC | Entropy | | NAE |
|---|---|---|---|---|---|
| | | | Cover image | Stego- image | |
| Fruits | 1 | 0.9996 | 7.488 | 7.496 | 0.009 |
| Peppers | 1 | 0.9997 | 7.573 | 7.603 | 0.012 |
| Boat | 1 | 0.9998 | 7.121 | 7.151 | 0.012 |
| House | 1 | 0.9998 | 5.756 | 6.630 | 0.007 |
| Lake | 1 | 0.9997 | 7.471 | 7.513 | 0.013 |
| Stream | 1 | 0.9991 | 7.702 | 7.719 | 0.020 |
| Livingroom | 1 | 0.9996 | 7.431 | 7.438 | 0.014 |
| Tulips | 1 | 0.9994 | 7.713 | 7.735 | 0.011 |
| Jetplane | 1 | 0.9998 | 6.716 | 6.795 | 0.008 |
| Cameraman | 1 | 0.9999 | 7.055 | 7.133 | 0.009 |
| **Average** | **1** | **0.9996** | **7.202** | **7.320** | **0.011** |

Moreover, in the SABMIS scheme, the measurement matrix $\Phi$, and the embedding/extraction algorithmic settings are considered as secret-keys, which are shared between the sender and the legitimate receiver. Even if the eavesdropper intercepting the stego-data becomes aware that the SABMIS scheme has been used to embed a secret image, he would not know these secret keys. Hence, we achieve increased security in our proposed system.

To justify this, we extract the secret image in two ways, *i.e.,* by using correct secret-keys and by using wrong secret-keys. Here, we embed only one secret image in a cover image although these experiments can be extended to the cases of embedding two, three or four secret images. Since the measurement matrix, which we use (random matrix having numbers with mean 0 and standard deviation 1) is one of the most commonly used measurement matrices and the eavesdropper might be able to guess it, we use this same measurement matrix while building wrong secret-keys. Here, we use the same dimension of this matrix as well, *i.e.,* $p3 \times p2$. In reality, the guessed matrix size would be different from the original matrix size, which would make the extraction task of the eavesdropper more difficult.

The algorithmic settings that we use will be completely unknown to the eavesdropper as above. These involve using a set of cover image coefficient indices where secret image coefficients are embedded ($p1$ and $p4$) and few constants ($\alpha = 0.01$, $\beta = 0.1$, $\gamma = 1$ and $c = 6$). While building wrong secret-keys, without changing the indices (*i.e.,* same $p1$ and $p4$), we take the common guess of one for all constants (*i.e.,* $\alpha = 1$, $\beta = 1$, $\gamma = 1$ and $c = 1$). In reality, the eavesdropper would not be able to correctly guess these indices as well, resulting in further challenges during extraction.

In Figs. 11A and 11B, we compare the 'Lake' secret image when extracted using correct and wrong secret-keys (from the 'Stream' stego-image), respectively. From this figure, we see that when using correct secret-keys, the visual distortion in the extracted secret image

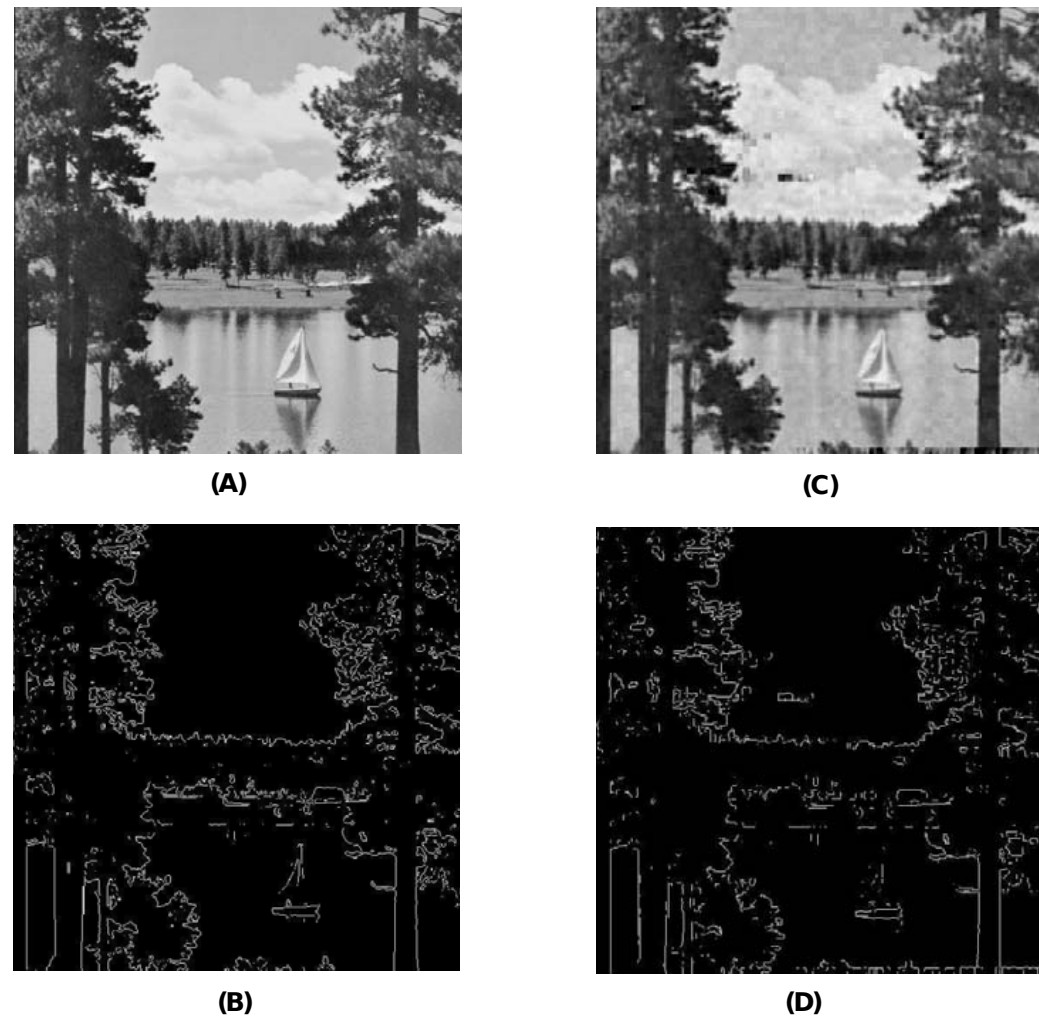

**Figure 10 Visual quality analysis between the 'Lake' original secret image and the 'Lake' extracted secret image (from the 'Stream' stego-image).** (A) 'Lake' original secret image, (B) Original secret image edge map, (C) 'Lake' extracted secret image, and (D) Extracted secret image edge map. Link: https://raw.githubusercontent.com/mohammadimtiazz/standard-test-images-for-Image-Processing/master/standard_test_images/lake.tif. Copyright: https://github.com/mohammadimtiazz/standard-test-images-for-Image-Processing/blob/master/LICENSE.

is negligible (as evident by comparing with Fig. 6E), and when using the wrong secret-keys, the distortion in the extracted secret image is very high (it is almost black).

Further, we numerically demonstrate that the correctly and wrongly extracted secret images are very different. We compute all the earlier discussed measures, *i.e.,* PSNR, MSSIM, NCC, Entropy, and NAE values between the correctly and wrongly extracted secret images (when all four secret images had been separately embedded in the ten cover images). The average values of all these metrics are given in Table 8. In this table, we observe that PSNR values are very low (recall over 30 dB are considered good). The MSSIM and

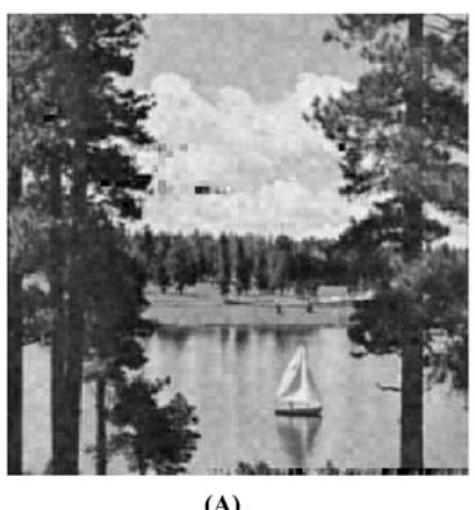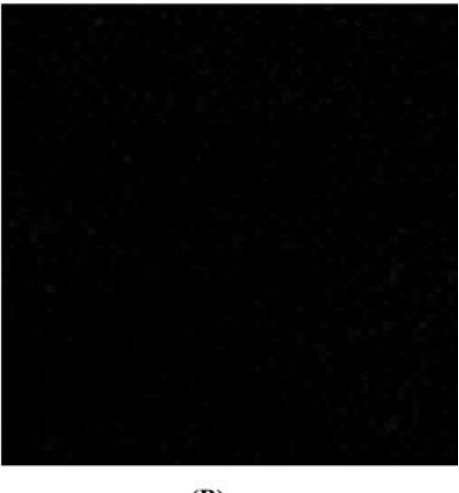

(A)                    (B)

**Figure 11** **Visual quality analysis between the 'Lake' extracted secret image when using correct and wrong secret-keys (from the 'Stream' stego-image).** (A) 'Lake' extracted secret image (when using correct secret-keys), and (B) 'Lake' extracted secret image (when using wrong secret-keys). Link: https://raw. githubusercontent.com/mohammadimtiazz/standard-test-images-for-Image-Processing/master/standard_ test_images/lake.tif. Copyright: https://github.com/mohammadimtiazz/standard-test-images-for-Image-Processing/blob/master/LICENSE.

**Table 8** Average PSNR, MSSIM, NCC, Entropy, and NAE value between the correctly and wrongly extracted secret images.

| Cover image | PSNR | MSSIM | NCC | Entropy | | NAE |
|---|---|---|---|---|---|---|
| | | | | Correctly extracted secret image | Wrongly extracted secret image | |
| Fruits | 6.032 | 0.0116 | 0.0037 | 7.188 | 1.409 | 0.9952 |
| Pepper | 5.767 | 0.0061 | 0.0034 | 7.604 | 1.419 | 0.9955 |
| Boat | 5.760 | 0.0070 | 0.0030 | 7.546 | 1.324 | 0.9959 |
| House | 5.767 | 0.0036 | 0.0015 | 7.533 | 0.897 | 0.9979 |
| Lake | 5.767 | 0.0083 | 0.0044 | 7.534 | 1.587 | 0.9942 |
| Stream | 5.835 | 0.0113 | 0.0071 | 7.542 | 1.974 | 0.9910 |
| Livingroom | 5.775 | 0.0078 | 0.0039 | 7.544 | 1.521 | 0.9948 |
| Tulips | 5.655 | 0.0162 | 0.0038 | 7.253 | 1.527 | 0.9948 |
| Airplane | 5.762 | 0.0074 | 0.00322 | 7.533 | 1.385 | 0.9956 |
| Cameraman | 5.780 | 0.0054 | 0.0025 | 7.531 | 1.151 | 0.9966 |
| **Average** | **5.790** | **0.0085** | **0.0037** | **7.481** | **1.419** | **0.9952** |

NCC values are close to 0. The entropy values of correctly and wrongly extracted secret images are far from each other. Finally, NAE values are close to 1. Hence, two images are substantially different from each other. Therefore, in the SABMIS scheme, a change in secret-keys will lead to a shift in the accuracy between the correctly and wrongly extracted secret images, in turn, making our scheme secure.

**Table 9  Timing data while embedding four secret images into different cover images.**

| Cover image | Run time of different stages of our SABMIS scheme (in Seconds) | | | |
| --- | --- | --- | --- | --- |
| | Hiding of secret images | Stego-image construction | Secret images extraction | Total time |
| Fruits | 8.92 | 74.67 | 12.78 | 96.37 |
| Pepper | 8.21 | 73.65 | 10.34 | 92.20 |
| Boat | 8.01 | 76.82 | 8.67 | 93.50 |
| House | 7.98 | 76.58 | 13.86 | 98.42 |
| Lake | 7.99 | 80.07 | 8.42 | 96.48 |
| Stream | 10.81 | 69.81 | 10.24 | 90.86 |
| Livingroom | 8.13 | 84.15 | 8.49 | 100.77 |
| Tulips | 8.68 | 81.34 | 9.13 | 99.15 |
| Airplane | 8.43 | 80.16 | 8.82 | 97.41 |
| Cameraman | 8.16 | 79.12 | 8.75 | 96.03 |
| **Average** | **8.38** | **77.34** | **9.83** | **95.55** |

## Timing data

The time taken by our SABMIS scheme is not of great importance here because all computations are done offline, whether it is hiding of secret images, stego-image construction, or the extraction of the secret images. However, for the sake of completeness, this data, while together hiding the four secret images in the ten cover images, is given in Table 9.

It is evident that the scheme is completely executed in a few minutes. Further, hiding and the extraction steps take about the same time (which they should because of similar steps), which is 10% of the total time. The most expensive step is stego-image construction, where the optimization problem is solved, which takes 80% of the total time.

## Application of our scheme on real-life data

In the two subsections below, we experiment on hiding mammograms and brain images (in cases where some loss is acceptable) in nondescript cover images. Sending these images safely across the internet is useful in breast cancer and brain related disease diagnosis, respectively. For the first case, we do not have reference steganographic data to compare against, while for the second case, we do have such data.

### Hiding mammograms

Here, we hide one through four mammograms (*Heath et al., 1998*; *Heath et al., 2001*) (see two in Figs. 12A and 12C) into all the cover images used in our experiments. These mammograms are freely available for research purposes. In Table 10, we present the embedding capacity and PSNR values from these experiments. As evident, we obtain good embedding capacity and average as well as maximum PSNR values. The other image comparison metrics turn out to be similar as well.

In Fig. 13, we present the visual comparison for 'Stream' as the cover image and the corresponding stego-image. We see that the cover and its corresponding stego-image are

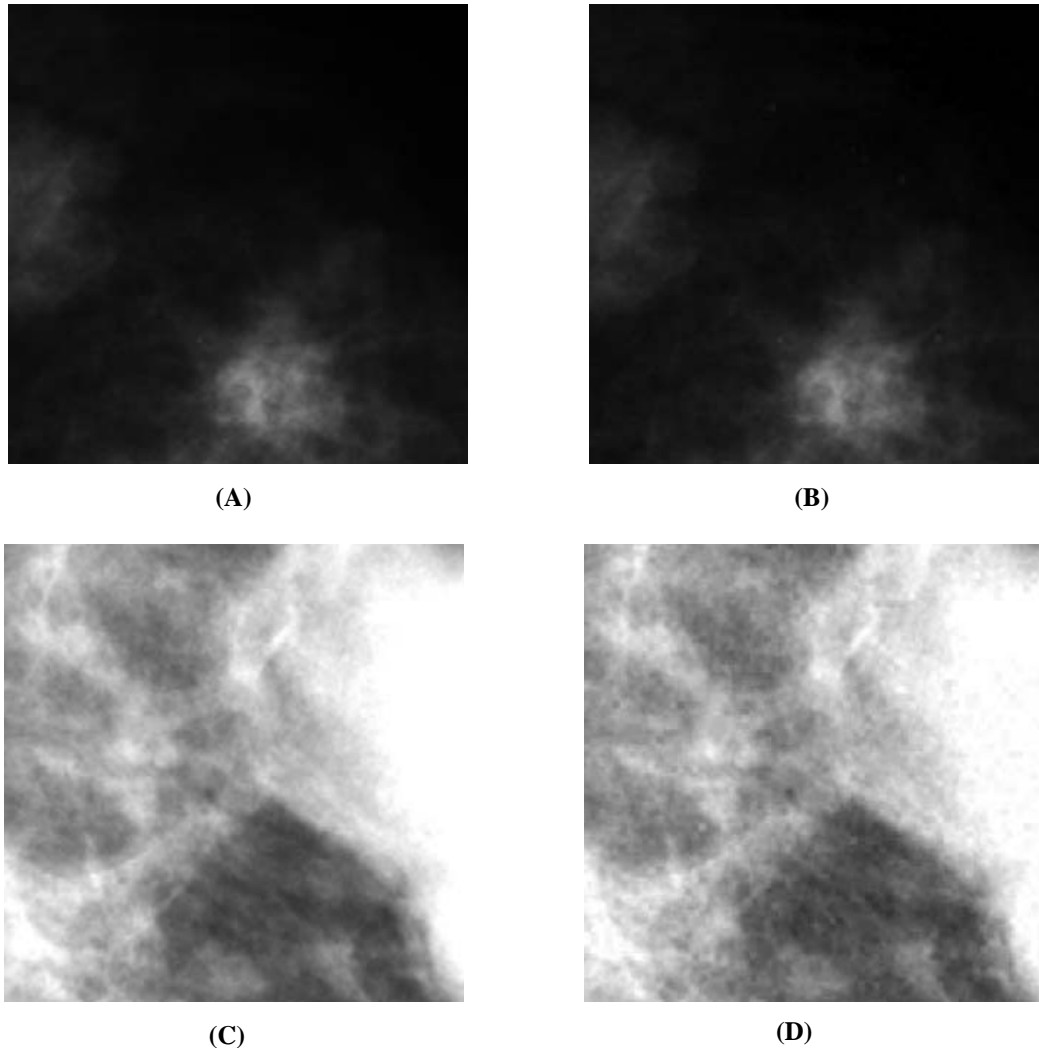

**Figure 12** **Visual quality analysis between the 'Mammogram' original secret image and the 'Mammogram' extracted secret image (from the 'Stream' stego-image).** (A) 'Mammogram' original secret image, (B) 'Mammogram' extracted secret image, (C) 'Mammogram' original secret image, and (D) 'Mammogram' extracted secret image. (A and C) source credits: Digital Database for Screening Mammography ©University of South Florida.

very similar. We get analogous results for the other images as well. We also check their edge maps (as discussed in section 'Stego-Image Quality Assessment') and obtain good results.

Next, we assess the quality of the extracted secret mammograms. In Figs. 12A and 12C, we show two original mammograms, and in Figs. 12B and 12D, we show the two respective extracted mammograms (from the 'Stream' stego-image). From these figures, we observe that there is very little distortion in the extracted mammograms. We get similar results for the other two mammograms as well.

**Table 10   Results of applicability of our scheme on real-life data (*i.e.,* mammograms).**

| No. of secret images | Steganography scheme | Type of secret image | Type of cover images | EC (in bpp) | (Avg. PSNR, No. of cover images) | Max. PSNR |
|---|---|---|---|---|---|---|
| 1 | SABMIS | Grayscale | Grayscale | 2 | (44.30, 10) | 49.41 |
| 2 | SABMIS | Grayscale | Grayscale | 4 | (35.54, 10) | 39.90 |
| 3 | SABMIS | Grayscale | Grayscale | 6 | (34.87, 10) | 39.10 |
| 4 | SABMIS | Grayscale | Grayscale | 8 | (34.32, 10) | 38.56 |

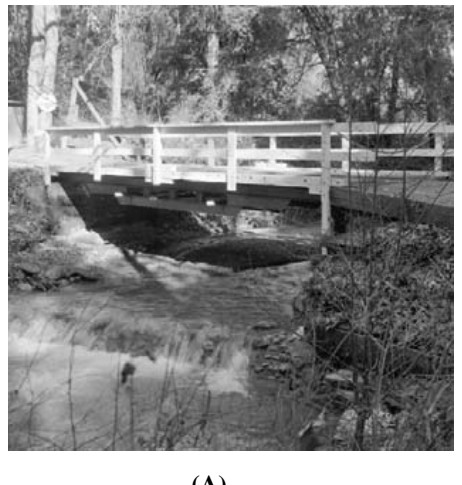 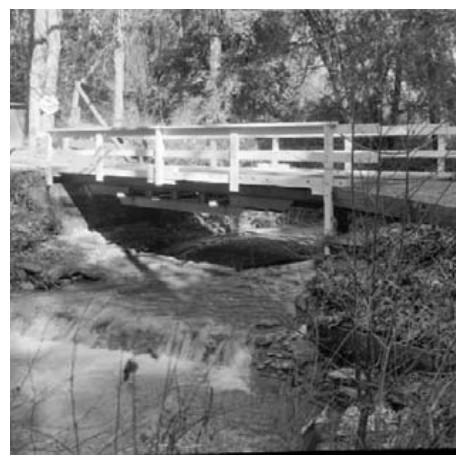

(A)            (B)

**Figure 13   Visual quality analysis between 'Stream' cover image (CI) and its corresponding stego-image (SI) when four mammograms are hidden.** (A) Cover Image, and (B) stego-image. Link: https://raw. githubusercontent.com/mohammadimtiazz/standard-test-images-for-Image-Processing/master/standard_ test_images/walkbridge.tif. Copyright: https://github.com/mohammadimtiazz/standard-test-images-for-Image-Processing/blob/master/LICENSE.

### Hiding brain images

*Arunkumar et al. (2019b)* hide a brain image into a cover image. Since the original brain image as used in *Arunkumar et al. (2019b)* is not publicly available, we work with an image that is quite similar to the image used in *Arunkumar et al. (2019b)*, and is available in free public domain with no copyright (see Fig. 14A) (*Rawpixel, 2022*; *Creative Commons, 2022*). By using SABMIS, we hide one through four copies of this image into all cover images (presented earlier), and compare with the results of *Arunkumar et al. (2019b)*.

This comparison is given in Table 11. As evident, we are not competitive with *Arunkumar et al. (2019b)* for the case of hiding one secret image (also discussed in 'Comparison with Past Work'). However, *Arunkumar et al. (2019b)*'s scheme can hide only one secret image while our scheme can hide multiple secret images. We observe that using SABMIS to hide four secret images in a cover image, we obtain a good embedding capacity of 8 bpp and a good average PSNR value of 33.56. The other image comparison metrics turn out to be similar as well.

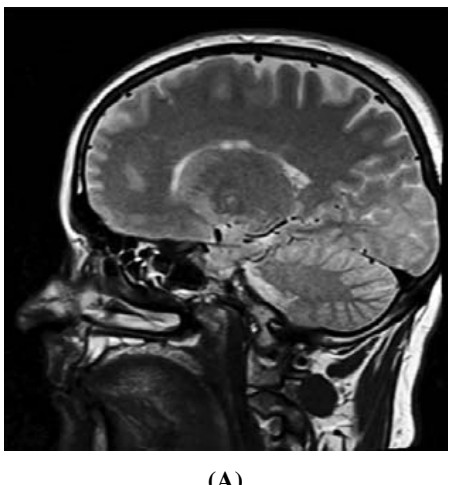

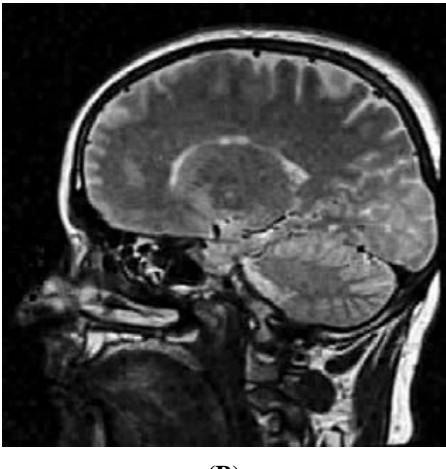

**(A)**                **(B)**

**Figure 14** **Visual quality analysis between the 'Brain' original secret image (*Bra, 2022a*; *Bra, 2022b*) and the 'Brain' extracted secret image (from the 'Stream' stego-image).** (A) 'Brain' original secret image, and (B) 'Brain' extracted secret image. Link: https://www.rawpixel.com/image/5939989/free-public-domain-cc0-photo. Copyright: https://creativecommons.org/publicdomain/zero/1.0/.

**Table 11** **Application of our scheme on real-life data (brain image), and its comparison with one scheme.**

| No. of secret images | Steganography scheme | Type of secret image | Type of cover images | EC (in bpp) | (Avg. PSNR, No. of cover images) | Max. PSNR |
|---|---|---|---|---|---|---|
| 1 | *Arunkumar et al. (2019b)* | Grayscale | Grayscale | 2 | (49.69, 8) | 50.15 |
| 1 | SABMIS | Grayscale | Grayscale | 2 | (41.54, 10) | 44.58 |
| 4 | SABMIS | Grayscale | Grayscale | 8 | (33.56, 10) | 37.74 |

As mentioned above, *Arunkumar et al. (2019b)* do not hide more than one secret image, and hence, we have no reference data to compare against in the rest of our results (quality of stego-image, quality of secret image, and resistant to steganographic attacks). In Fig. 15, we present the visual comparison of 'Stream' as the cover image and the corresponding stego-image while hiding four copies of this brain image. As evident, the cover and its corresponding stego-image are very similar. We get analogous results for the other cover images as well. We also check their edge maps (as discussed in section 'Stego-image Quality Assessment st2') and obtain good results.

In Fig. 14, we show the original brain secret image and one of the extracted brain images (from the 'Stream' stego-image). From these figures, we observe that when compared with the original secret image, the quality of the extracted secret image is good. Finally, like (*Arunkumar et al., 2019b*), our scheme is inherently resistant to steganographic attacks. Our design makes our scheme more robust.

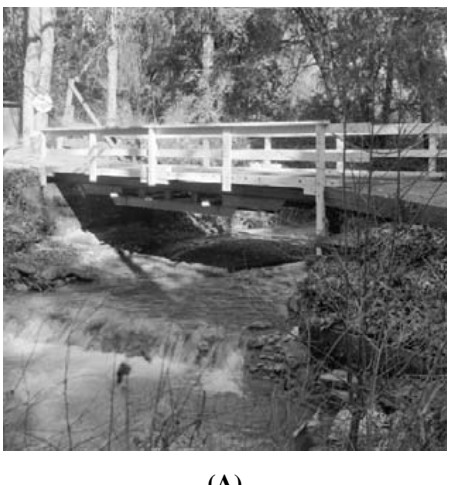 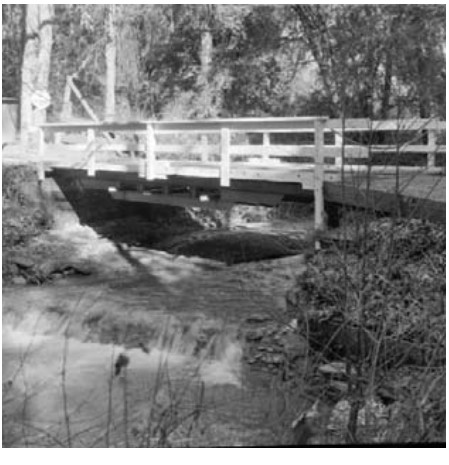

(A)                                                              (B)

**Figure 15** **Visual quality analysis between 'Stream' cover image (CI) and its corresponding stego-image (SI) when four copies of brain medical images are hidden.** (A) Cover image, and (B) stego-image. Link: https://raw.githubusercontent.com/mohammadimtiazz/standard-test-images-for-Image-Processing/master/standard_test_images/walkbridge.tif. Copyright: https://github.com/mohammadimtiazz/standard-test-images-for-Image-Processing/blob/master/LICENSE.

## CONCLUSIONS AND FUTURE WORK

In image steganography, the challenges are increasing the embedding capacity of the scheme, maintaining the quality of the stego-image as well as extracted secret image, and ensuring that the scheme is resistant to steganographic attacks. We propose SABMIS, a blind multi-image steganography scheme for securing secret images in cover images to substantially overcome these challenges. All our images are grayscale, which is a hard problem.

Our proposed SABMIS consists of many novel features to tackle the above challenges. This includes a novel embedding rule that embeds the secret image sparse coefficients into oversampled cover image sparse coefficients in a staggered manner; a transformed LASSO formulation of the underlying optimization problem to construct the stego-image, which is eventually solved by ADMM; and finally, the reverse of our unique embedding rule resulting in an extraction rule.

We perform exhaustive experiments to demonstrate that our scheme overcomes all the challenges of image steganography as discussed above. We focus on embedding multiple secret images. The embedding capacity of SABMIS for the case of embedding two and three secret images is the best in the published literature (three times and six times than the existing best, respectively). While embedding four secret images, our embedding capacity is slightly lower than *Hu (2006)* (about $\frac{2}{3}^{rd}$) but we do substantially better in overcoming the other challenges.

The quality of our stego-images (when compared with the corresponding cover images) and our extracted secret images (when compared with the corresponding original secret images) are the best among the existing literature (over 30 dB of PSNR values). SABMIS is

intrinsically as well as designed to be resistant to steganographic attacks (because transform based and algorithmic settings, respectively), making it one of the most secure schemes among the existing ones.

Additionally, we show that SABMIS can be applied in a very little amount of time, and also demonstrate SABMIS's successful application on the real-life problem of securely sending medical images over the internet.

Next, we discuss the future work in this context. First is further *improving* our algorithm. As mentioned earlier, our SABMIS scheme has multiple novel components. Although in Appendix, 'Sensitivity of our scheme with respect to the novel components', we perform sensitivity analysis of SABMIS with respect to one such component (oversampling), a more detailed analysis is part of future work. In future, we plan to find improved values of parameters $\alpha, \beta, \gamma$, etc. used in the embedding and the extraction aspects of SABMIS. Further, our scheme may give poor results when embedding more than four secret images (see Appendix, 'A possible scenario where our scheme is not the best'). Hence, exploring this aspect is also part of our future work.

Second is *extending* our scheme to embed images into videos because the amount of information that may be hidden in an image is limited. Third is *adapting* our scheme for real-life applications. Although in this article, we discuss use of SABMIS for securing mammograms and brain images while transmitting them over the internet, extensive experiments for this are part of our future work. Another related application is safely sharing biometric data of people over the internet. We plan to explore this aspect in future as well.

## APPENDIX. SOME STEGANOGRAPHY SCHEMES FOR HIDING BINARY SECRET DATA

As discussed in the introduction, our focus is on hiding images into an image, and the images can be binary, grayscale, or color. Hiding binary data into images is a separate problem because the evaluation metrics for hiding images and binary data are completely different. However, for the sake of completeness, in Table A1, we summarize some existing works that discuss hiding of binary data into images. These papers are sorted in the decreasing order of date of publishing.

## APPENDIX. A SMALL NUMERICAL EXAMPLE OF OUR EMBEDDING PROCESS

Our embedding process for a small example (with $2 \times 2$ blocks for both the secret and cover images) is shown in Fig. A1. In the experiments, we show the results of hiding/ embedding up to four secret images in a cover image. However, for the sake of simplicity, here, we show the case of hiding one secret image into a cover image.

Table A1 **Some steganography schemes for hiding binary secret data into an image.** All cover images are colored below.

| Reference | Technique |
|---|---|
| *AlKhodaidi & Gutub (2021)* | LSB (Least Significant Bits) |
| *Al-Shaarani & Gutub (2021a)* | LSB and DWT (Discrete Wavelet Transform) |
| *Al-Shaarani & Gutub (2021b)* | LSB and DWT |
| *Hureib & Gutub (2020)* | LSB |
| *Gutub & Al-Ghamdi (2020)* | LSB |
| *Almutairi, Gutub & Al-Ghamdi (2019)* | LSB |
| *Gutub & Al-Ghamdi (2019)* | A modified version of LSB |
| *Alanizy et al. (2018)* | LSB |
| *Gutub & Al-Juaid (2018)* | LSB |
| *Parvez & Gutub (2011)* | A modified version of LSB |
| *Gutub (2010)* | A modified version of LSB |

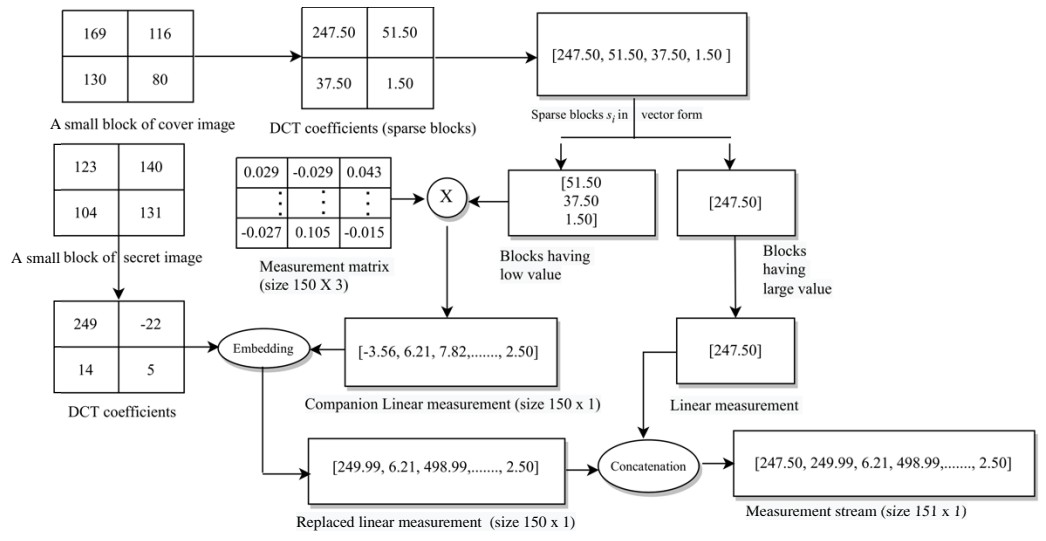

Figure A1 **A small numerical example of secret image embedding.**

# APPENDIX. A SMALL NUMERICAL EXAMPLE OF OUR STEGO-IMAGE CONSTRUCTION PROCESS

Our stego-image construction process, from the stego-data obtained from Fig. A1, is shown in Fig. A2.

# APPENDIX. SENSITIVITY OF OUR SCHEME WITH RESPECT TO THE NOVEL COMPONENTS

Here, we demonstrate that when we omit or restrict a particular component of our steganography scheme, then how it affects the overall performance. As discussed earlier, the

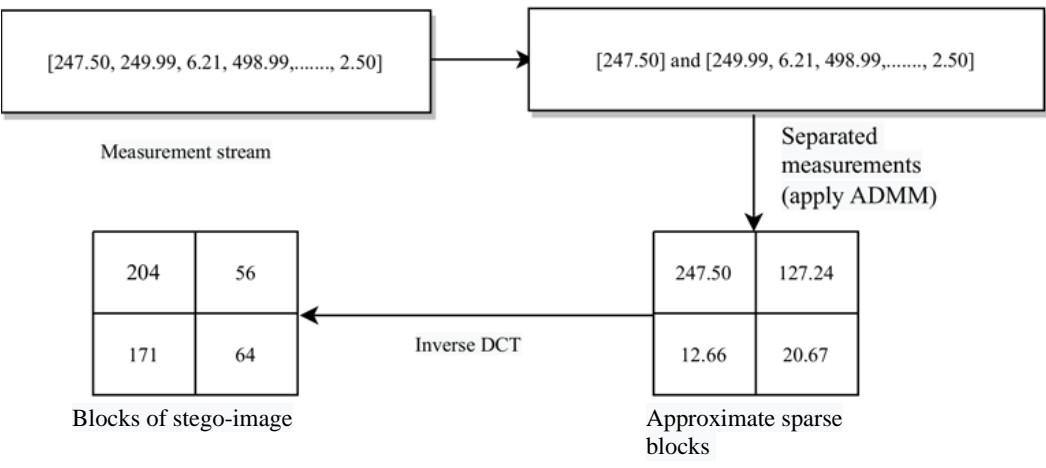

**Figure A2  A small numerical example of stego-image construction.**

novel components of SABMIS are: the oversampling of the cover image sparse coefficients and hiding secret image sparse data into them in a staggered way (our embedding rule); using ADMM to solve the LASSO formulation of the underlying minimization problem for stego-image construction; and the extraction of the secret images by the extraction rule (which is the reverse of the embedding rule).

Without loss of generality, we restrict the oversampling component and show its effects on the performance.[9] As mentioned in the experimental result section (*i.e.,* in section 'Experimental Results'), the size of the measurement matrix $\Phi$ is $p_3 \times p_2$ with $p_3 > p_2$. Earlier, we took $p_3 = 50 \times p_2$. Here, we take $p_3 = 2 \times p_2$, *i.e.,* we restrict this oversampling. In Fig. A3, we show the stego-image PSNR values for the case of hiding one, two, three, and four secret images with this restricted oversampling in SABMIS. Comparing this figure with Fig. 9 (hiding one to four secret images with original oversampling in SABMIS), we observe that the PSNR values reduce substantially. Hence, the novel component of oversampling of our SABMIS scheme greatly affects the overall performance.[10]

## APPENDIX. A POSSIBLE SCENARIO WHERE OUR SCHEME IS NOT THE BEST

Here, we give a possible scenario where our scheme does not give the best results. We hide six (instead of four) secret images using our proposed steganography scheme and check all the evaluation metrics discussed earlier. The secret images chosen are shown in Figs. 6A, 6B, 6D, 6E, 6F, and 6J.

We achieve up to 12 bpp embedding capacity. Visually, both the cover image and the stego-image are almost identical (see Fig. A4). While looking at the numerical measures, we achieve an average PSNR value of 34.39 dB, average MSSIM value close to 0.9991, average NCC value of 0.9981, nearly same entropy of the cover image and the stego-image,

[9]Since we design our embedding rule in such a way that we always need the number of linear measurements larger than the number of sparse coefficients, we could not completely omit this oversampling.

[10]We obtain similar results with other comparison metrics as well.

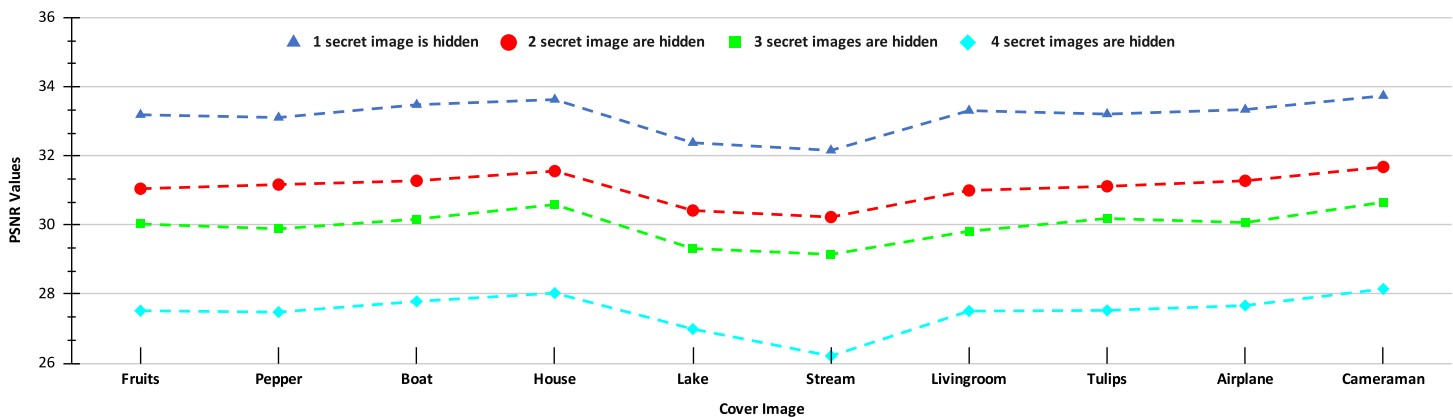

**Figure A3  PSNR values of the stego-images when different numbers of images are hidden in the ten cover images (with restricted oversampling in SABMIS).**

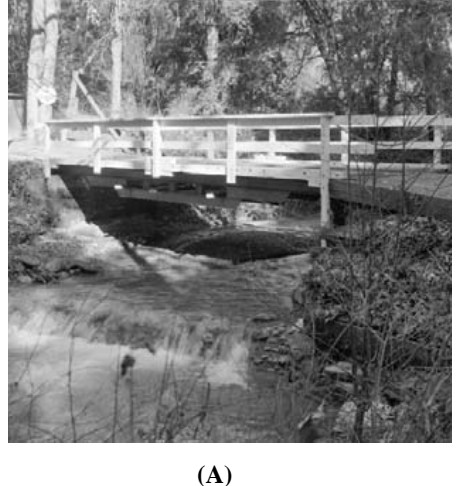

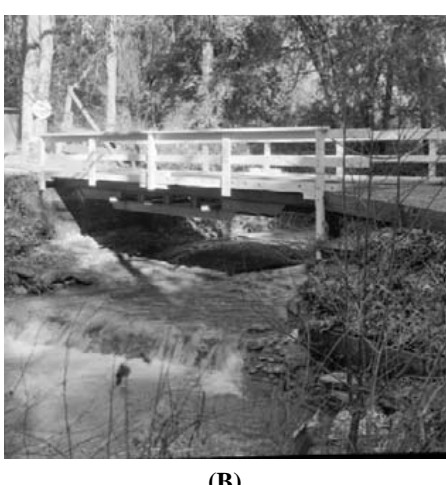

**(A)**  **(B)**

**Figure A4  Visual quality analysis between 'Stream' cover image (CI) and its corresponding stego-image (SI), when hiding six secret images.** (A) Cover image, and (B) stego-image. Link: https://raw.githubusercontent.com/mohammadimtiazz/standard-test-images-for-Image-Processing/master/standard_test_images/walkbridge.tif. Copyright: https://github.com/mohammadimtiazz/standard-test-images-for-Image-Processing/blob/master/LICENSE.

and average NAE value close to 0. All these values further indicate that the stego-image is very similar to its corresponding cover image. However, the original secret image and the extracted secret image are very different (see Fig. A5). Hence, we observe that when we try to hide more than four secret images using our scheme, the quality of the extracted secret images degrades.

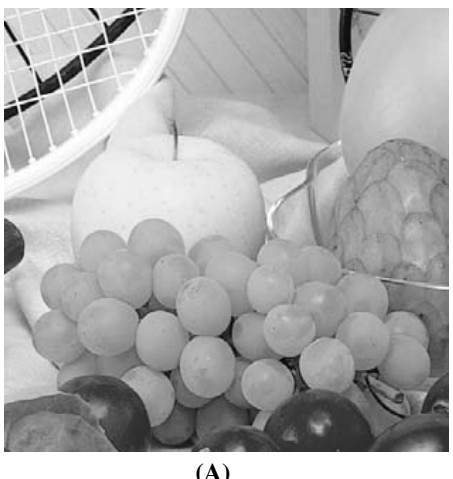
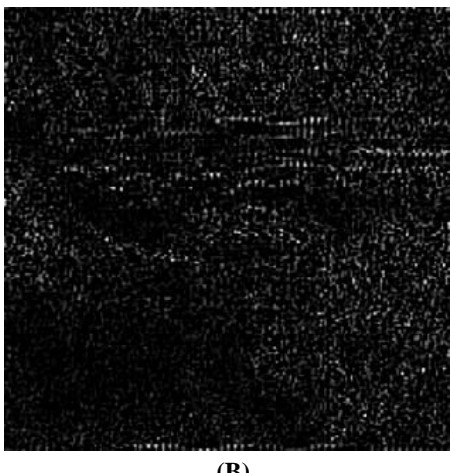

(A)                                                    (B)

**Figure A5** **Visual quality analysis between the 'Fruits' original secret image and the 'Fruits' extracted secret image (from the 'Stream' stego-image, when hiding six secret images).** (A) 'Fruits' original secret image, and (B) 'Fruits' extracted secret image. Link: https://github.com/mohammadimtiazz/standard-test-images-for-Image- Processing/blob/master/standard_test_images/fruits.png. Copyright: https://github.com/mohammadimtiazz/standard-test-images-for-Image-Processing/blob/master/LICENSE.

### Funding

This work was supported by DAAD (Germany) under the project 'A new passage to India' between Indian Institute of Technology Indore and the Leibniz Universität Hannover. The funders had no role in study design, data collection and analysis, decision to publish, or preparation of the manuscript.

### Grant Disclosures

The following grant information was disclosed by the authors:
DAAD (Germany) under the project 'A new passage to India' between Indian Institute of Technology Indore and the Leibniz Universität Hannover.

### Competing Interests

The authors declare there are no competing interests.

### Author Contributions

- Rohit Agrawal conceived and designed the experiments, performed the experiments, analyzed the data, performed the computation work, prepared figures and/or tables, authored or reviewed drafts of the article, and approved the final draft.
- Kapil Ahuja conceived and designed the experiments, performed the experiments, analyzed the data, authored or reviewed drafts of the article, and approved the final draft.

- Marc C. Steinbach analyzed the data, authored or reviewed drafts of the article, and approved the final draft.
- Thomas Wick analyzed the data, authored or reviewed drafts of the article, and approved the final draft.

## Data Availability

The Images to be used for experiments are provided in the Supplementary File 1 (ImagesFiles.zip).

The matlab scripts for all the experiments are provided in the Supplementary File 2 (MatlabFiles.zip).

All the instructions to execute the experiments and descriptions of all the parameters and matlab files are provided in the Readme file attached as Supplementary File 3 (Readme.txt).

## Supplemental Information

Supplemental information for this article can be found online at http://dx.doi.org/10.7717/peerj-cs.1080#supplemental-information.

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
