# Peer review of "SABMIS: sparse approximation based blind multi-image steganography scheme"

_PeerJ Computer Science, doi:10.7717/peerj-cs.1080_

## Round 0.1 · original submission · Major Revisions

The reviewers identified some merits of the paper but a revision is needed. Please note that I do not expect you to cite any recommended reference unless crucial. Please provide a detailed response letter. Thanks.

Reviewer 1 ·

Basic reporting

This paper presents an interesting image steganography system of hiding grayscale secret images into another grayscale cover image. The embedding of secret data is done to enhance embedding capacity while preserving the visual quality of the stego-image to be ensuring that the stego-image is resistant to steganographic attacks. The tests quantitatively and qualitatively experimental results claimed efficacy of the proposed method over others. The evaluation and comparison results provided interesting remarks very much pioneering, but needs to be improved and justified compared to others, in order to be ready as publication, as noted within the following points that have to be fulfilled:
--The abstract needs to be briefed a bit stressing more on specific key contribution and originality trying to attract the reader to select the paper to read and refer to as well as get motivated toward the work to continue research in similar direction.
--Give more elaboration on the real need for utilizing this proposal. What is wrong in the normal other stego hiding methods requiring this kind of complex security research. Try to support your explanation via real-life examples.

Experimental design

--The implementation needs more elaboration in different ways. You need to add a numerical example to make the reader grasp the proposed model and its applicability thoroughly. The example must be simple arranged in a convincing manner to allow further research improvement to build upon.
-- Try showing more possible scenarios with details where your work can give less efficient remarks.
-- Figure 8 is strange representing the average NAE value relating between the four original secret images and their corresponding extracted secret images when using correct and wrong secret-keys in a very consistent manner. This needs checking and further explanation ?

Validity of the findings

--The work lack reasoning and comparisons. The work needs many reasoning and comparison to others proofing its real applicability. It also needs elaboration proofing fairness in the results and comparison, especially testing the schemes on the different images. It needs some indication with more justification of these results. Explain more the opposite observations feedback of the proposed work vs. others, i.e. as showing different results of evaluation with other the results.
-- The image stego research needs it work coverage to be completed by involving some supplementary related image hiding researches as well as counting-based secret sharing. Accordingly, the work must involve the stego current topics, in order to be ready:
== Increasing Participants Using Counting-Based Secret Sharing via Involving Matrices and Practical Steganography
== Enhancing Medical Data Security via Combining Elliptic Curve Cryptography with 1-LSB and 2-LSB Image Steganography
== Refining image steganography distribution for higher security multimedia counting-based secret-sharing
== Hiding Shares by Multimedia Image Steganography for Optimized Counting-Based Secret Sharing
== Securing Matrix Counting-Based Secret-Sharing Involving Crypto Steganography
== Image Steganography to Facilitate Online Students Account System
== 3-Layer PC Text Security via Combining Compression, AES Cryptography 2LSB Image Steganography
== Image Based Steganography to Facilitate Improving Counting-Based Secret Sharing
== Multi-Bits Stego-System For Hiding Text in Multimedia Images Based on User Security Priority
== Vibrant Color Image Steganography using Channel Differences and Secret Data Distribution
== Pixel Indicator Technique for RGB Image Steganography

Additional comments

-- Conclusion needs reconsideration. It needs to highlight more of the research main contributions with some brief indications and numerical improvement percentages to keep with the reader. Also, the conclusion needs to present some additional specific ideas of open research and future work for researchers to build upon for further advancements.

·

Basic reporting

In this paper the authors propose a novel blind image steganography scheme using sparse approximation and Alternating Direction Method of Multipliers (ADMM) to embed upto four secret images in single cover image with block wise DCT sparsification method. The proposed method can achieve state-of-the-art performance and can perform hiding multiple images in a single cover image. The manuscript is clearly written in professional language.

Experimental design

Correctness: In general correctness is up to the mark. However it is claimed that there is no other steganography method available in literature which can embed four secret images into a single secret image. The authors should search the literature thoroughly since such a scheme to hide four secret images into one cover image was available in 2006 itself. (Kindly refer: Yu-Chen Hu, Multiple Images Embedding Scheme Based on Moment Preserving Block Truncation Coding, Fundamenta Informaticae 73 (2006) 373–387 373 IOS Press).

Validity of the findings

Strengths: In this paper the authors have generated the stego image using the Alternating Direction Method of Multipliers (ADMM) to solve the Least Absolute Shrinkage and Selection Operator (LASSO) formulation which is quite interesting. A sufficient number of statistical measures are used to evaluate the performance of the image steganography scheme developed in the proposed work. The paper represents a well defined and detailed research in the field of image steganography using DCT. The method for analyzing is interesting.

Weaknesses: The authors should incorporate execution time analysis to support the strengths claimed in the manuscript. In addition the authors have mentioned that the payload capacity is higher in the proposed work and at the same time it is questionable that there is a limitation of having the cover image size double of the secret image size.

Additional comments

The article is well written and can be accepted with few corrections as detailed below.
1. Execution time analysis should be added.
2. Comparative analysis should be elaborated since there are existing methods available in literature which can be compared with the proposed work.
3. Justification on the claimed payload capacity is required.

·

Basic reporting

The manuscript proposes a blind multi-image steganography scheme utilizing sparse approximation and novel embedding rule claiming to improve the embedding capacity and enhance security too. The manuscript has a well-written introduction and very recent relevant literature has all been cited and authors could relate their work with these recent works and put it in proper perspective. All the figures and data reported through the analysis are optimum.

At some places, the discussion appears disconnected and that, I believe, is due to inefficient use of language. Therefore the language part could have been improved a bit.

Experimental design

The manuscript reports an original piece of work by the authors and attempts a contemporary research problem that provides a better solution. The objectives and motivation behind the proposed work and its strategy are well explained. The algorithm and its steps of embedding the secret image, generating the stego-image and extraction of the secret images have been clearly explained with sufficient details. Authors could highlight their own contribution while discussing their algorithm in line with the recent works too.

Validity of the findings

For assessing the security of the proposed algorithm, the authors have used the images from well-known image databases and analysed all essential parameters like embedding capacity, stego-image quality and metrics like PSNR, MSSIM, NCC, NAE etc and the results prove the superiority of the proposed algorithm.

However, in the security analysis part, the authors claim that their algorithm is secure against the steganographic attacks just with the help of some arguments and through analysis by computing the NAE for the correct and wrong set of keys which appear a little incomplete or do not justify their claim fully. This part requires some improvement with some more technical details and supporting analysis.

---

## Round 0.2 · accepted · Accept

The paper can be accepted. Congratulations.

Reviewer 1 ·

Basic reporting

All comments have been incorporated in satisfactory level. The paper is revised perfect. Recommend ACCEPTANCE.

Experimental design

All comments have been incorporated in satisfactory level. The paper is revised perfect. Recommend ACCEPTANCE.

Validity of the findings

All comments have been incorporated in satisfactory level. The paper is revised perfect. Recommend ACCEPTANCE.

Additional comments

All comments have been incorporated in satisfactory level. The paper is revised perfect. Recommend ACCEPTANCE.

·

Basic reporting

In this paper the authors propose a novel blind image steganography scheme using sparse approximation and Alternating Direction Method of Multipliers (ADMM) to embed up to four secret images in single cover image with block wise DCT sparsification method. The proposed method can achieve state-of-the-art performance and can perform hiding multiple images in a single cover image.

In general correctness is up to the mark and suggested changes in earlier review have been taken care of.

Experimental design

The proposed work is analyzed in different possible ways and compared with the recent research too. Sharing of multiple secret images is achieved successfully using transform domain embedding method. The suggested references have been cited and analyzed up to the mark.

Validity of the findings

Conclusion is well elaborated highlighting the strengths as well as weaknesses of the proposed work. A sufficient number of statistical measures are used to evaluate the performance of the image steganography scheme developed in the proposed work. The paper represents a well defined and detailed research in the field of image steganography using DCT. The method for analyzing is interesting.

Additional comments

The article is well written and can be accepted as the following suggestions have been incorporated in the revised manuscript.
4. 1. Execution time analysis is added.
2. Comparative analysis is elaborated with the existing
methods available in literature and is compared with the
proposed work.
3. Justification on the claimed payload capacity is incorporated.

---

## Author Rebuttal · Round 0.2

# Answers to Reviewer 1

Rohit Agrawal    Kapil Ahuja    Marc C. Steinbach
Thomas Wick

July 14, 2022

We thank the reviewer for spending time and giving us a detailed feedback. This has helped us greatly improve our results, and subsequently, the quality of our paper. All issues have been addressed.

For answers to specific points, please see below. All updates to the manuscript are colored in blue for your easy reference. The edits corresponding to the other reviewers are colored in brown, and purple color. We have done some minor improvements in our writing throughout the manuscript but these are not color coded so as to not distract from the main updates.

We thank the editor for giving us valuable extensions in this time of need.

# Comments by Reviewer

Comment 1. Basic reporting

   This paper presents an interesting image steganography system of hiding grayscale secret images into another grayscale cover image. The embedding of secret data is done to enhance embedding capacity while preserving the visual quality of the stego-image to be ensuring that the stego-image is resistant to steganographic attacks. The tests quantitatively and qualitatively experimental results claimed efficacy of the proposed method over others. The evaluation and comparison results provided interesting remarks very much pioneering, but needs to be improved and justified compared to others, in order to be ready as publication, as noted within the following points that have to be fulfilled:

1.
- The abstract needs to be briefed a bit stressing more on specific key contribution and originality trying to attract the reader to select the paper to read and refer to as well as get motivated toward the work to continue research in similar direction.

- Give more elaboration on the real need for utilizing this proposal. What is wrong in the normal other stego hiding methods requiring this kind of complex security research. Try to support your explanation via real-life examples.

**Response to the Comment:**

Thank you for these very useful suggestions.

- We have now completely revamped the abstract on page i. In paragraph 2, we have highlighted our originality more clearly (please see the blue colored sentences). Paragraphs 3, 4, and 5 are completely new that emphasize the worth of this research by comparing with existing best.

- We have immensely improved the manuscript on the real need of our sophisticated scheme. Kindly see the following places:

   (a) Abstract on page i: Described as above.

   (b) Introduction (Section 1): From lines 87–98, we had earlier motivated the need of such an approach, however, in the next paragraphs (lines 99–108), we have now motivated it further with a telediagnosis example. In the subsequent paragraph (lines 109–112), we have now emphasized the novelty of our scheme better.

(c) Comparison with Past Work (Section 1.1) starting on page v: Here, we have now exhaustively compared (via table as well as descriptively) on why the past works do not efficiently address the problem we are solving. In the earlier manuscript, this sub-section was very brief (located in the Results section) while now in the new manuscript it is very exhaustive (located in the Introduction section).

(d) Application on Real-life Data (Section 3.6) starting on page xxv: In this new subsection, we have now demonstrated the application of our scheme on securely transmitting mammograms and brain images (part of telediagnosis).

(e) Conclusions (Section 4) starting on page xxvii: This mirrors the improvements done in the abstract. Please see the first 5 paragraphs.

Comment 2. Experimental design

2.
- The implementation needs more elaboration in different ways. You need to add a numerical example to make the reader grasp the proposed model and its applicability thoroughly. The example must be simple arranged in a convincing manner to allow further research improvement to build upon.

- Try showing more possible scenarios with details where your work can give less efficient remarks.

- Figure 8 is strange representing the average NAE value relating between the four original secret images and their corresponding extracted secret images when using correct and wrong secret-keys in a very consistent manner. This needs checking and further explanation ?

**Response to the Comment:**

We thank the reviewer for these very useful useful suggestions. The above concern have been addressed as below.

- We have improved the exposition of our implementation in three ways. Firstly, we have now explained the indices used in our embedding rule better (please see lines 324–343).

    Secondly, we have added block diagrams for all the three components of our algorithms. That is, Figures 3, 4, and 5 (on pages xii, xiii, and xv, respectively) demonstrate concisely our embedding process, stego-image construction, and extraction process, respectively.

Thirdly, using a numerical example, we have reemphasized the workings of the embedding and the stego-image construction process. The example for embedding process is given in Appendix B (on page xxx), and for stego-image construction is given in Appendix C (on page xxx). Our extraction process (on page xiii) is exactly the reverse of our embedding process (discuss in the manuscript).

All the above mentioned figures and appendices have been appropriately referred in Proposed Approach (Section 2) starting on page vii.

- We have now given two possible weaknesses of our algorithm. Firstly, in Appendix D (on page xxx), we have shown that our results are sensitive to the amount of oversampling done in the embedding process. Hence, doing the "correct" amount of oversampling is important.

  Secondly, in Appendix E (on page xxxii), we have now shown that our scheme may give poor results when embedding more than four secret images. Exploring both these aspect is part of our future work, and hence, has been referred in Conclusions and Future Work (Section 4) starting on page xxvii.

- The way we were representing the data in Figure 8 of the original manuscript did not depict the change in the NAE values (due to rounding at $5^{th}$ place of decimal). Sorry about this. We have now removed this figure and added numerical results in a new table (Table 8 on page xxiv) that avoids this confusion.

  Moreover, based on the suggestion of another reviewer, we have now compared the correctly extracted secret image and the wrongly extracted secret image using all the measures presented in the paper. That is, PSNR, MSSIM, NCC, Entropy, and, as earlier, NAE. Please see Table 8 (on page xxiv) and lines 630–640.

Comment 3. Validity of the findings

3. • The work lack reasoning and comparisons. The work needs many reasoning and comparison to others proofing its real applicability. It also needs elaboration proofing fairness in the results and comparison, especially testing the schemes on the different images. It needs some indication with more justification of these results. Explain more the opposite observations feedback of the proposed work vs. others, i.e. as showing different results of evaluation with other the results.

• The image stego research needs it work coverage to be completed by involving some supplementary related image hiding researches as well as counting-based secret sharing. Accordingly, the work must involve the stego current topics, in order to be ready:

(a) Increasing Participants Using Counting-Based Secret Sharing via Involving Matrices and Practical Steganography

(b) Enhancing Medical Data Security via Combining Elliptic Curve Cryptography with 1-LSB and 2-LSB Image Steganography

(c) Refining image steganography distribution for higher security multimedia counting-based secret-sharing

(d) Hiding Shares by Multimedia Image Steganography for Optimized Counting-Based Secret Sharing

(e) Securing Matrix Counting-Based Secret-Sharing Involving Crypto Steganography

(f) Image Steganography to Facilitate Online Students Account System

(g) 3-Layer PC Text Security via Combining Compression, AES Cryptography 2LSB Image Steganography

(h) Image Based Steganography to Facilitate Improving Counting-Based Secret Sharing

(i) Multi-Bits Stego-System For Hiding Text in Multimedia Images Based on User Security Priority

(j) Vibrant Color Image Steganography using Channel Differences and Secret Data Distribution

(k) Pixel Indicator Technique for RGB Image Steganography

**Response to the Comment:**

- Yes, in the earlier manuscript, the comparison with existing work was sparsely done. Thanks to your very useful comment. We have now comprehensively addressed this aspect (reasoning, comparison, proofing, justification, real need, and opposite observations).

  Please see our complete answer to your comment 1. Also, please see the second bullet of our answer to your comment 2.

- Thank you for suggesting this. As discussed in the Introduction of our manuscript, our focus is on embedding images into an image, and an image can be binary, grayscale, or color. The papers suggested by you focus on embedding binary data into images, which is a different track and not our focus. However, to make our paper more comprehensive (thanks a lot again for suggesting), we have now summarized these papers in Appendix A (on page xxx) and referred this appendix in the Introduction on line 56.

Comment 4. Additional comments

4.
- Conclusion needs reconsideration. It needs to highlight more of the research main contributions with some brief indications and numerical improvement percentages to keep with the reader. Also, the conclusion needs to present some additional specific ideas of open research and future work for researchers to build upon for further advancements.

**Response to the Comment:**
Many thanks for your valuable suggestion. This substantially helped us improve the Conclusions and Future Work (Section 4) starting on page xxvii. Main research contributions (along with numerical data) have been expanded upon in the first 5 paragraphs of this section.

We have now discussed three main future directions (in great detail) in the last 2 paragraphs of this section as well.

Comment 5. Our note

5.

**Response to the Comment:**
Finally, based upon the comments of the other reviewers, we have additionally made one more major positive update to the manuscript. That is, we have added timing data of SABMIS in Section 3.5 starting on page xxiv.

# References

[1] Al-Shaarani, F., and Gutub, A. (2021). Increasing participants using counting-based secret sharing via involving matrices and practical steganography. Arabian Journal for Science and Engineering, In Press.

[2] Hureib, E. S. B., and Gutub, A. A. (2020). Enhancing medical data security via combining elliptic curve cryptography with 1-LSB and 2-LSB image steganography. IJCSNS, 20(12):232.

[3] AlKhodaidi, T., and Gutub, A. (2021). Refining image steganography distribution for higher security multimedia counting-based secret-sharing. Multimedia Tools and Applications, 80(1):1143–1173.

[4] Gutub, A., and Al-Ghamdi, M. (2020). Hiding shares by multimedia image steganography for optimized counting-based secret sharing. Multimedia Tools and Applications, 79(11):7951–7985.

[5] Al-Shaarani, F., and Gutub, A. (2021). Securing matrix counting-based secret-sharing involving crypto steganography. Journal of King Saud University-Computer and Information Sciences, In Press.

[6] Almutairi, S., Gutub, A., and Al-Ghamdi, M. (2019). Image steganography to facilitate online students account system. Rev. Bus. Technol. Res, 16(2):43–49.

[7] Alanizy, N., Alanizy, A., Baghoza, N., AlGhamdi, M., and Gutub, A. (2018). 3-layer PC text security via combining compression, AES cryptography 2LSB image steganography. Journal of Research in Engineering and Applied Sciences (JREAS), 3(4):118-124.

[8] Gutub, A., and Al-Ghamdi, M. (2019). Image based steganography to facilitate improving counting-based secret sharing. 3D Research, 10(1):6.

[9] Gutub, A., and Al-Juaid, N. (2018). Multi-bits stego-system for hiding text in multimedia images based on user security priority. Journal of computer hardware engineering, 1(2):1-9.

[10] Parvez, M. T., and Gutub, A. A. A. (2011). Vibrant color image steganography using channel differences and secret data distribution. Kuwait J Sci Eng, 38(1B):127–142.

[11] Gutub, A. A. A. (2010). Pixel indicator technique for RGB image steganography. Journal of emerging technologies in web intelligence, 2(1):56–64.

# Answers to Reviewer 2

Rohit Agrawal     Kapil Ahuja     Marc C. Steinbach
Thomas Wick

July 14, 2022

We thank the reviewer for spending time and giving us a detailed feedback. This has helped us greatly improve our results, and subsequently, the quality of our paper. All issues have been addressed.

For answers to specific points, please see below. All updates to the manuscript are colored in brown for your easy reference. The edits corresponding to the other reviewers are colored in blue, and purple color. We have done some minor improvements in our writing throughout the manuscript but these are not color coded so as to not distract from the main updates.

## Comments by Reviewer

1.
> Comment 1. Basic reporting.
> In this paper the authors propose a novel blind image steganography scheme using sparse approximation and Alternating Direction Method of Multipliers (ADMM) to embed upto four secret images in single cover image with block wise DCT sparsification method. The proposed method can achieve state-of-the-art performance and can perform hiding multiple images in a single cover image. The manuscript is clearly written in professional language.

**Response to the Comment:**

We are thankful to the reviewer for appreciating our work.

2.

> Comment 2. Experimental design
> Correctness: In general correctness is up to the mark. However it is claimed that there is no other steganography method available in literature which can embed four secret images into a single secret image. The authors should search the literature thoroughly since such a scheme to hide four secret images into one cover image was available in 2006 itself. (Kindly refer: Yu-Chen Hu, Multiple Images Embedding Scheme Based on Moment Preserving Block Truncation Coding, Fundamenta Informaticae 73 (2006) 373–387 373 IOS Press).

**Response to the Comment:**

We are very thankful to the reviewer for pointing us to [1]. We again thoroughly searched the literature and found one more relevant paper [2]. [1] and [2] have proposed steganography schemes to hide four secret images into one cover image using the spatial domain approach, which is not inherently resistant to steganographic attacks [3, 4] [1]. In comparison, in our scheme we hide four secret images using the transform based approach, which is intrinsically resistant to steganographics attacks.

Since no one besides us has attempted embedding three or four secret images while using the transform based approach, we have now performed comparisons as below.

- For the case of embedding one or two secret images, we have now compared against the best schemes that embed one or two secret images using the transform based approach.

- For the case of embedding three or four secret images, we have now compared with the best schemes that embed three or four secret images using the spatial-domain based approach, i.e., [5] and [2], respectively.

  In all cases, we have now comprehensively and clearly demonstrated the superiority of our algorithm. Please see

  (a) Updated Abstract (paragraphs three to five) on page i.
  (b) Comparison with Past Work (Section 1.1) starting on page v.
  (c) Updated Conclusions (paragraphs three to five) on pages xxviii to xxix.

As earlier, all changes as compared to earlier submitted manuscript are now colored.
* * *
[1] As a side note, we have now listed these two papers in Table 1 on page ii of our updated manuscript.

Comment 3. Validity of the findings

Strengths: In this paper the authors have generated the stego image using the Alternating Direction Method of Multipliers (ADMM) to solve the Least Absolute Shrinkage and Selection Operator (LASSO) formulation which is quite interesting. A sufficient number of statistical measures are used to evaluate the performance of the image steganography scheme developed in the proposed work. The paper represents a well defined and detailed research in the field of image steganography using DCT. The method for analyzing is interesting.

Weaknesses: The authors should incorporate execution time analysis to support the strengths claimed in the manuscript. In addition the authors have mentioned that the payload capacity is higher in the proposed work and at the same time it is questionable that there is a limitation of having the cover image size double of the secret image size.

3.

**Response to the Comment:**

We are very thankful to the reviewer for highlighting the strengths of our work. We also thank the reviewer for highlighting the weakness of the earlier version of the manuscript. We have now addressed both these concerns in this current version as below.

- We have now added a new section on timing data where we have extensively given results while embedding four secret images in all the cover images chosen for our experiments. Here, we have shown the amount of time taken by the three components of our algorithm. Overall, SABMIS executes in few minutes. Please see Timing Data (Section 3.5) on page xxiv of the updated manuscript.

- Many thanks for this deep insight on the payload capacity. We completely agree that our better embedding capacity is due to the design choice of making the length and the width of cover image double of the length and the width of secret image, respectively. Our biggest strength is that while maintaining this high embedding capacity,

  (a) we achieve very good quality stego-image (over 30 dB PSNR values and other metrics)

  (b) we obtain the extracted secret images to be of almost the same quality as the original secret images (which most competing works do not demonstrate)

  (c) we have designed a highly secure steganography scheme (because the base is transform based with oversampling & embedding rule built on top of it)

We have now changed our narration to highlight this aspect. Please see

(a) Abstract on page i: lines 23 to 43.

(b) Introduction (Section 1): lines 87 to 98.

(c) Experimental Results (Section 3): lines 470 to 477.

> **Comment 4. Additional comments**
> The article is well written and can be accepted with few corrections as detailed below.
>
> 1. Execution time analysis should be added.
>
> 2. Comparative analysis should be elaborated since there are existing methods available in literature which can be compared with the proposed work.
>
> 3. Justification on the claimed payload capacity is required.

4.

**Response to the Comment:**

Many thanks again for your feedback.

- Execution time analysis, as mentioned in the second paragraph of the answer to your comment 3 above, has been done now.

- Comparative analysis, as mentioned in the last paragraph of the answer to your comment 2 above, has now been done exhaustively.

- Payload capacity justification, as mentioned in the last two paragraphs of the answer to your comment 3 above, has been done now as well.

> **Comment 5. Our note**

5.

**Response to the Comment:**
Finally, based upon the suggestions of the other reviewers, we have performed the following major improvements to the manuscript.

(a) Highlighted the novelty of our algorithm better:

    – Abstract lines 15 to 21

    – Introduction (Section 1) lines 109 to 112.

    – Conclusion and Future Work (Section 4) lines 703 to 715.

(b) Added real life examples:

# References

[1] Hu, Y. C. (2006). Multiple images embedding scheme based on moment preserving block truncation coding. Fundamenta Informaticae, 73(3): 373–387.

[2] Manujala, G. R. and Danti, A. (2015). Embedding multiple images in a single image using Bit Plane Complexity Segmentation (BPCS) steganography. Asian Journal of Mathematics and Computer Research, 2(3): 136–142.

[3] Artiemjew, P. and Aleksandra, K. M. (2020). Indiscernibility mask key for image steganography. Computers, 9(2): 38.

[4] Hassaballah, M., Hameed, M. A., Awad, A. I. and Muhammad, K.(2021). A novel image steganography method for industrial internet of things security. IEEE Transactions on Industrial Informatics, 17(11): 7743–7751.

[5] Guttikonda, P., Cherukuri, H. and Mundukur, N. B. (2018). Hiding encrypted multiple secret images in a cover image. In Proceedings of International Conference on Computational Intelligence and Data Engineering, pages 95–104. Springer.

# Answers to Reviewer 3

Rohit Agrawal     Kapil Ahuja     Marc C. Steinbach

Thomas Wick

July 14, 2022

We thank the reviewer for spending time and giving us a detailed feedback. This has helped us greatly improve our results, and subsequently, the quality of our paper. All issues have been addressed.

For answers to specific points, please see below. All updates to the manuscript are colored in purple for your easy reference. The edits corresponding to the other reviewers are colored in blue, and brown color. We have done some minor improvements in our writing throughout the manuscript but these are not color coded so as to not distract from the main updates.

We thank the editor for giving us valuable extensions in this time of need.

## Comments by Reviewer

1.

> Comment 1. Basic reporting
>
> The manuscript proposes a blind multi-image steganography scheme utilizing sparse approximation and novel embedding rule claiming to improve the embedding capacity and enhance security too. The manuscript has a well-written introduction and very recent relevant literature has all been cited and authors could relate their work with these recent works and put it in proper perspective. All the figures and data reported through the analysis are optimum.
>
> At some places, the discussion appears disconnected and that, I believe, is due to inefficient use of language. Therefore the language part could have been improved a bit.

**Response to the Comment:**

We are grateful to the reviewer for appreciating our work. As advised, we have improved the language throughout the manuscript and performed many positive updates as below.

(a) Added real life examples:
  – Introduction (Section 1) lines 99 to 108.
  – Application of Our Scheme on Real-life Data (Section 3.6) starting on page xxv.

(b) Improved the exposition of our scheme:
  – Proposed Approach (Section 2) lines 324 to 343.
  – new figures 3, 4, & 5 on pages xii, xiii, & xv, respectively
  – new Appendices B & C on page xxx.

(c) We have added timing data of SABMIS in Section 3.5 on pages xxiv to xxv.

(d) Substantially expanded upon the future work:
  – Conclusions and Future Work lines 732 to 746.
  – new Appendices D & E on pages xxx to xxxii.

2.

> Comment 2. Experimental design
>
> The manuscript reports an original piece of work by the authors and attempts a contemporary research problem that provides a better solution. The objectives and motivation behind the proposed work and its strategy are well explained. The algorithm and its steps of embedding the secret image, generating the stego-image and extraction of the secret images have been clearly explained with sufficient details. Authors could highlight their own contribution while discussing their algorithm in line with the recent works too.

**Response to the Comment:**

Many thanks for highlighting the strengths of our manuscript. We agree that exposition of contributions and comparison was slightly weak in the earlier submitted manuscript. We have now improved these aspects as below.

(a) Abstract on page i: We have now completely revamped the abstract. In paragraph 2, we have highlighted our originality more clearly (please see the blue colored sentences). Paragraphs 3, 4, and 5 are completely new that emphasize the worth of this research by comparing with existing best.

(b) Introduction (Section 1): From lines 87–98, we had earlier motivated the need of such an approach, however, in the next paragraphs (lines 99–108), we have now motivated it further with a telediagnosis example. In the subsequent paragraph (lines 109–112), we have now emphasized the novelty of our scheme better.

(c) Comparison with Past Work (Section 1.1) starting on page v: Here, we have now exhaustively compared (via table as well as descriptively) on why the past works do not efficiently address the problem we are solving. In the earlier manuscript, this sub-section was very brief (located in the Results section) while now in the new manuscript it is very exhaustive (located in the Introduction section).

(d) Application on Real-life Data (Section 3.6) starting on page xxv: In this new subsection, we have now demonstrated the application of our scheme on securely transmitting mammograms and brain images (part of telediagnosis).

(e) Conclusions (Section 4) starting on page xxvii: This mirrors the improvements done in the abstract. Please see the first 5 paragraphs.

Comment 3. Validity of the findings

For assessing the security of the proposed algorithm, the authors have used the images from well-known image databases and analysed all essential parameters like embedding capacity, stego-image quality and metrics like PSNR, MSSIM, NCC, NAE etc and the results prove the superiority of the proposed algorithm.

However, in the security analysis part, the authors claim that their algorithm is secure against the steganographic attacks just with the help of some arguments and through analysis by computing the NAE for the correct and wrong set of keys which appear a little incomplete or do not justify their claim fully. This part requires some improvement with some more technical details and supporting analysis.

3.

**Response to the Comment:**

We are again very thankful to the reviewer for this feedback. We have now exhaustively compared the correctly and wrongly extracted secret images not just visually and by using NAE values but by all the comparative measures presented in this manuscript. That is PSNR, MSSIM, NCC, Entropy, and as earlier, NAE. Please see lines 630 to 640 and accompanying Table 8 on page xxiv.